# Leveraging Side Information with Deep Learning for Linear Inverse Problems: Applications to MR Image Reconstruction

## Abstract

Reducing the time it takes to acquire a Magnetic Resonance Imaging (MRI) scan is an important problem in healthcare, as it can improve patient care and reduce costs. One way to achieve this is by acquiring only a fraction of the frequency space data and reconstructing diagnostic-quality images from it. This problem can be formulated as a linear inverse problem (LIP), where the forward operator, which maps the structure of the imaged object to the acquired frequency space data, can become rank-deficient or exhibit many small singular values. This leads to ambiguities in the reconstruction process, where multiple images (most of them non-diagnostic) can map to the same set of acquired data. To resolve these ambiguities, it is essential to leverage domain knowledge and, whenever possible, exploit additional context (a.k.a., *relevant side information*) when solving the LIP. We present a novel, end-to-end trainable deep learning-based method, called **T**rust-**G**uided **V**ariational **N**etwork (**TGVN**), that reliably incorporates side information into LIPs to eliminate undesirable solutions from the *ambiguous space* of the forward operator, while remaining faithful to the acquired data. We demonstrate its effectiveness through applications in multi-coil, multi-contrast MR image reconstruction, where incomplete or low-quality measurements from one contrast are used as side information to reconstruct a high-quality image of another contrast from heavily under-sampled data. Its robustness is validated by reconstructing images from different contrasts across different anatomies and field strengths. Compared to a set of baselines that also use side information, our method reconstructs high-quality images in the presence of heretofore challenging levels of under-sampling, thereby speeding up the acquisition drastically while providing protection against hallucinations. Our approach is also versatile enough to incorporate many different types of side information into any LIP.

## 1 Introduction

Magnetic Resonance Imaging (MRI) is a mainstay of medical diagnostic imaging, thanks to its flexibility, its rich information content, and its excellent soft-tissue contrast. An MR scanner collects measurements in frequency space (a.k.a., $k$-space) that encode the body's response to applied electromagnetic fields, with multiple receiver coils capturing distinct views modulated by their individual sensitivities. These measurements are then used to reconstruct high-fidelity diagnostic quality images. The problem of image reconstruction from multi-coil $k$-space data can be formulated as a Linear Inverse Problem (LIP), where the objective is to deduce an accurate representation of structures in an object of interest (i.e., an image) from the observed measurements. The term "linear" refers to the linear relationship between the observed measurements and the object of interest, and is defined by a known process called the *forward operator*.

Despite MRI's superior diagnostic capabilities, it is comparatively time-consuming and costly, which limits its overall accessibility. Reducing the time it takes to acquire an MR scan is an important practical problem that can improve patient care, limit patient discomfort, reduce costs, and improve accessibility of this imaging modality. One way to reduce scan time is to acquire a smaller number of $k$-space measurements. The challenge then becomes how to reconstruct high-quality images from limited data. Undersampling in $k$-space renders the underlying LIP ill-posed or ill-

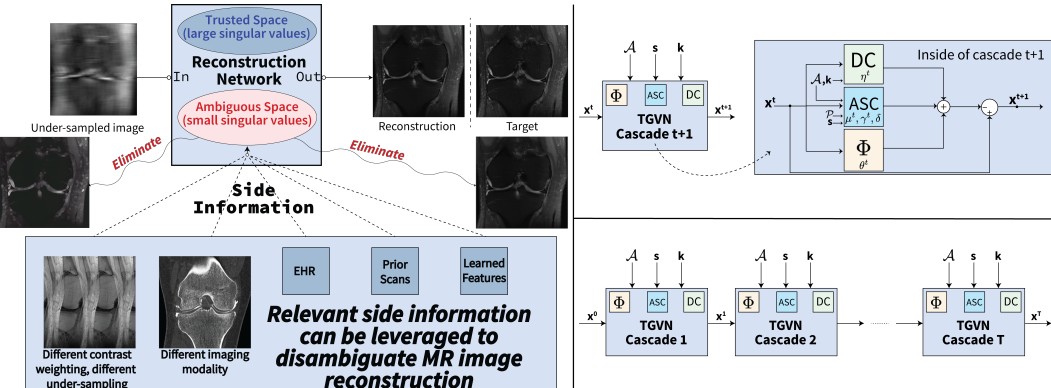

Figure 1: **Reconstruction with side information, a TGVN cascade, and a full TGVN.** Left: Illustration of TGVN in high-level. Upper right: A TGVN cascade and its inside operations consisting of data consistency (DC), ambiguous space consistency (ASC) and refinement ($\Phi$). The novelty of our work is the ASC module which exploits the side information ($\mathbf{s}$) and projection onto the ambiguous space ($\mathcal{P}$) to further disambiguate the reconstruction. Lower right: A full TGVN consisting of $T$ cascades connected in series.

conditioned, because the forward operator that captures the relationship between the image and the observed measurements becomes either rank deficient or ill-conditioned, leading to ambiguities in the reconstruction process: multiple solutions (most of them clinically infeasible) can map to the same set of acquired data.

Researchers have proposed several solutions, including the total variation approach (Block et al., 2007; Ma et al., 2008), low-rank penalty methods (Lingala et al., 2011; Shin et al., 2014), compressed-sensing-based methods (Lustig et al., 2007; 2008), and priors learned from exemplary data or directly from the measurements themselves (Ravishankar & Bresler, 2010; Lingala & Jacob, 2013; Caballero et al., 2014). Recent advances in machine learning, particularly through the development of deep learning techniques, have markedly improved the ability to tackle these ill-posed or ill-conditioned problems. Notable early examples include the ADMM-Net approach Yang et al. (2016), the Variational Network (VarNet) approach (Hammernik et al., 2018; Sriram et al., 2021), the Model-Based Deep Learning (MoDL) approach (Aggarwal et al., 2019), the ISTA-Net approach (Zhang & Ghanem, 2019) and the FISTA-Net approach (Xiang et al., 2021), all of which integrate traditional optimization techniques with deep neural networks to achieve robust and efficient solutions in high-dimensional spaces. More recently, researchers have proposed generative models for reconstructing high-quality images from incomplete data (Bora et al., 2017; Chung & Ye, 2022; Song et al., 2022; Peng et al., 2022), and transformer-based methods (Huang et al., 2022; Guo et al., 2023), and a rapidly-expanding portfolio of deep-learning-based image reconstruction methods is currently under development.

Another approach to limiting degenerate solutions to such ill-posed or ill-conditioned LIPs involves leveraging additional contextual information (a.k.a., *relevant side information*), often as regularizers or constraints in the optimization problem's objective function (Weizman et al., 2015; 2016; Ehrhardt & Betcke, 2016; Song et al., 2019; Zhou & Zhou, 2020; Bian et al., 2022; Lei et al., 2023b). The nature of such side information is problem-dependent, and in many real-world scenarios it is readily available. Relevant side information can take multiple forms, including images, text, or other types of structured data. In MR image reconstruction, for instance, the side information could be data associated with prior scans of the patient. It could also be data gathered during the same scan, such as images obtained using an imaging pulse sequence with a different underlying contrast from the target pulse sequence. In more general settings, the side information need not be derived from the same imaging modality, nor does it need to be image-based; it could be textual (e.g., clinical notes and medical history), audio, or even encoded features or representations learned from other related tasks. We note that reconstruction with different contrast side information, also known as *conditional reconstruction*, refers to reconstructing only the target contrast while exploiting information from other contrasts. This approach differs from both single-contrast and joint multi-contrast reconstruction, and is the scope of our work.

**Contributions:** In this work, we propose a novel end-to-end trainable deep learning method that reliably integrates side information to solve LIPs. Our method, called the **T**rust-**G**uided **V**ariational **N**etwork (**TGVN**), uses the side information to disambiguate the subspace spanned by the trailing right singular vectors of the forward operator of an LIP, i.e., the right singular vectors corresponding to small singular values. Specifically, we introduce a learnable squared Euclidean distance constraint, termed the *ambiguous space consistency* constraint, into the regularized least-squares reconstruction formulation of the LIP to eliminate undesirable solutions from the ambiguous space of the forward operator. This *ambiguous space consistency* constraint can be seamlessly integrated into any deep unrolled network. Our approach can be trained end-to-end with full supervision to maximize a similarity metric between the reconstructed and the ground truth image, requiring minimal modifications to integrate the constraint. By incorporating additional contextual data, our approach effectively reduces the ambiguities inherent in inverse problems, leading to more accurate and reliable solutions even when the measurements are exceedingly sparse. We demonstrate the effectiveness of our method in the challenging domain of multi-coil, multi-contrast MR image reconstruction, where incomplete or low-quality measurements from complementary contrast weighting are used as side information to reconstruct images with a different target contrast from drastically small quantities of $k$-space measurements (of the orders of $20\times$ low sampling rates) across different anatomies and field strengths. Compared to recently proposed machine learning-based solutions, our method leverages side information efficiently by focusing on the solution space creating ambiguities while maintaining consistency with the acquired measurements and achieves statistically significant improvements in reconstruction performance, highlighting the advantage of integrating additional context. To summarize:

- We propose a novel method called the **T**rust-**G**uided **V**ariational **N**etwork (**TGVN**) that leverages side information to reliably solve an ill-posed or ill-conditioned LIP.

- We demonstrate the effectiveness of TGVN in multi-coil multi-contrast MR image reconstruction, using incomplete or low-quality measurements from complementary contrast weighting as side information.

- We demonstrate the robustness of our method by showing its efficacy for different contrasts across multiple anatomies and multiple field strengths.

- We show that TGVN leverages side information more efficiently than recent ML-based solutions, achieving statistically significant improvements in image reconstruction performance and pushing the boundaries of current techniques in medical imaging and beyond.

## 2 BACKGROUND

### 2.1 MULTI-COIL MR IMAGE ACQUISITION

In MR imaging, measurements are acquired in the spatial frequency (a.k.a., $k$-space) domain, and the measurements are related to the estimated MR image through the linear forward operator $\mathcal{A}$. These measurements may be grouped into a complex-valued vector $\widetilde{\mathbf{k}}$, and the elements of $\widetilde{\mathbf{k}}$ represent Fourier coefficients of the structure of the continuous object being imaged. Specifically, we define a discrete estimated MR image $\mathbf{x}$, such that $\widetilde{\mathbf{k}} = \mathcal{F}(\mathbf{x}) + \epsilon$, where $\epsilon$ is complex Gaussian noise and $\mathcal{F}$ denotes the Fourier transform operator. The vector $\mathbf{x} \in \mathbb{C}^{MN}$ is a complex vector of size $MN$, where $M$ and $N$ are pixel dimensions of the two-dimensional image being sought.

In parallel imaging (PI), the scanner captures multiple views of the anatomy modulated by the sensitivities $S_i$ of the receiver coils, which can be represented by diagonal matrices $S_i \in \mathbb{C}^{MN \times MN}$. In this case the relationship becomes: $\widetilde{\mathbf{k}}_i = \mathcal{F}(S_i\mathbf{x}) + \epsilon_i$, for each $i \in \{1, 2 \ldots, N_c\}$, where $N_c$ denotes the number of coils. To simplify notation, we aggregate the k-space data from all coils into a single tensor $\widetilde{\mathbf{k}} = (\widetilde{\mathbf{k}}_1, \ldots, \widetilde{\mathbf{k}}_{N_c})$ and define the *expand* operator $(\mathcal{E})$ which maps the complex image to multi-coil k-space. That is, $\mathcal{E} : \mathbf{x} \mapsto (\mathcal{F}(S_1\mathbf{x}), \ldots, \mathcal{F}(S_{N_c}\mathbf{x}))$.

To accelerate MR acquisition, fewer k-space samples are acquired, which we denote by a binary diagonal mask matrix $\mathcal{M} \in \{0, 1\}^{MN \times MN}$, of size $MN \times MN$, whose diagonal element is set to $1$ only if the corresponding measurement was acquired. Otherwise, it is set to $0$. Thus, the under-sampled k-space can be denoted as $\mathbf{k} \triangleq \mathcal{M}\widetilde{\mathbf{k}} = \left(\mathcal{M}\widetilde{\mathbf{k}}_1, \ldots, \mathcal{M}\widetilde{\mathbf{k}}_{N_c}\right)$, and the forward

operator $\mathcal{A}$—mapping the underlying image to the under-sampled and noisy k-space—in multi-coil MR image acquisition is equal to $\mathcal{M} \circ \mathcal{E}$. That is,

$$\mathbf{k} = \mathcal{A}\mathbf{x} + \epsilon' = (\mathcal{M} \circ \mathcal{E})\,\mathbf{x} + \epsilon'. \tag{1}$$

## 2.2 Deep Learning for Parallel MR Image Reconstruction

Given the forward operator $\mathcal{A}$ and the $k$-space data $\mathbf{k}$, estimating $\mathbf{x}$ is considered a *well-posed* problem if it meets the following three criteria (called the Hadamard conditions): 1) existence of a solution, 2) uniqueness of the solution, and 3) stability of the solution (Hansen, 2010). Accelerated parallel MR image reconstruction, however, like most real-word problems, is either *ill-posed*, failing to meet one or more of these criteria, or *ill-conditioned*, with small errors in the measurements leading to much larger errors in our image estimate $\mathbf{x}$. This is because the sparse set of measurements $\mathbf{k}$ makes the above system of equations (equation 1) either under-determined, with a potentially infinite set of solutions, or ill-conditioned, with a large yet finite condition number. When the measurement noise is Gaussian, the maximum likelihood estimate of a solution to equation 1 is given by $\widehat{\mathbf{x}} = \arg\min_{\mathbf{x}} \frac{1}{2}\|\mathcal{A}\mathbf{x} - \mathbf{k}\|_2^2$. To address its ill-posed or ill-conditioned nature, the inverse problem is reformulated to impose additional constraints or requirements on the solution. By incorporating appropriate additional constraints, such as regularization, one can derive a reliable approximate solution. More formally, let $\Psi(\cdot)$ denote a regularization function that imposes certain constraints on the possible solutions $\mathbf{x}$. Then the optimization problem, equation 2.2, can be modified as:

$$\widehat{\mathbf{x}} = \arg\min_{\mathbf{x}} \frac{1}{2}\|\mathcal{A}\mathbf{x} - \mathbf{k}\|_2^2 + \Psi(\mathbf{x}). \tag{2}$$

In deep-learning based unrolled networks, such as End-to-end Variational Network (E2E-VarNet) (Sriram et al., 2021), one learns a regularization function from the training data to maximize a desired similarity metric between the reconstructed image $\widehat{\mathbf{x}}$ and the ground truth. Specifically, E2E-VarNet starts with an initial estimate $\mathbf{x}^0$ of the solution to $\mathcal{A}\mathbf{x} = \mathbf{k}$, and uses a gradient descent scheme with respect to $\mathbf{x}$ for a fixed number of steps $T$ to iteratively refine its estimate and solve equation 2. Furthermore, it replaces the gradient of the regularization function $\Psi(\mathbf{x})$ with a neural network $\Phi$, parametrized by $\theta^t$ at each iteration $t$. More formally, E2E-VarNet executes the following sequence of steps for a total of $T$ iterations, starting with $\mathbf{x}^0 = \mathcal{A}^H\mathbf{k}$:

$$\mathbf{x}^{t+1} = \mathbf{x}^t - \eta^t \mathcal{A}^H\left(\mathcal{A}\mathbf{x}^t - \mathbf{k}\right) - \Phi\left(\mathbf{x}^t; \theta^t\right), \tag{3}$$

where $\mathcal{A}^H = \mathcal{E}^H \circ \mathcal{M}$ is the Hermitian adjoint of $\mathcal{A}$. It is worth mentioning that the second term on the right hand side is usually referred to as *data consistency*, as it guides $\mathbf{x}$ to be maximally consistent with the acquired measurements. At the end of iteration $T$, we obtain $\mathbf{x}^T$ parameterized by $\Theta = \{\theta^0, \dots, \theta^{T-1}, \eta^0, \dots, \eta^{T-1}\}$. Assuming access to ground truth $\mathbf{x}^*$, parameters $\Theta$ are learned in a supervised manner to maximize a desired similarity between $\mathbf{x}^T$ and $\mathbf{x}^*$.

## 3 Related Work: Side Information in MR Image Reconstruction

This section summarizes how prior work has leveraged side information in MR image reconstruction. While side information can take various forms, most studies have focused on complementary contrast information. The task involves reconstructing the target contrast using information from other contrasts, differing from both single-contrast and joint multi-contrast reconstruction.

**Initial Attempts.** The use of side information in medical image reconstruction dates back to at least the 1990s. Fessler et al. (1992) demonstrated tomographic image reconstruction based on a weighted Gibbs penalty, where the weights are determined by anatomical boundaries in high-resolution MR images. Gindi et al. (1993) proposed a Bayesian method whereby maximum a posteriori (MAP) estimates of PET and SPECT images may be reconstructed with the aid of prior information derived from registered anatomical MR images of the same slice. Some of the earlier attempts also utilized handcrafted priors (Haldar et al., 2008; Wu et al., 2011; Peng et al., 2011; Bilgic et al., 2011; Du & Lam, 2012; Huang et al., 2014; Qu et al., 2014; Li et al., 2015; Weizman et al., 2015; 2016; Ehrhardt & Betcke, 2016). Later, dictionary-learning-based methods were proposed (Gungor et al., 2017; Song et al., 2019; Lei et al., 2023a).

**End-to-end Methods.** More recently, multiple authors have proposed end-to-end deep learning-based models that leverage side information for MR image reconstruction. Specifically, Xiang et al.

Figure 2: **Example showing the use of side information to reconstruct an image from heavily under-sampled $k$-space data.** **Left:** Coronal PDFS image (main information) from $14\times$ under-sampled $k$-space data. **Middle:** Coronal PD image (side information) from $3\times$ equispaced under-sampled data. **Right:** Reconstructed coronal PDFS image along with the ground truth target image. Since PD- and PDFS-weighted scans share certain features, despite their distinct contrast and under-sampling patterns, using PD as side information to guide PDFS reconstruction is beneficial.

(2018; 2019) proposed combining T1-weighted images and under-sampled T2-weighted images to reconstruct fully-sampled T2-weighted images using a Dense-U-net model. Zhou & Zhou (2020) introduced a Dilated Residual Dense Network (DuDoRNet) for dual domain restorations from under-sampled MRI data to simultaneously recover $k$-space and images. Feng et al. (2022) developed a multi-modal transformer ('MTrans') for accelerated MR imaging for transferring multi-scale features from the target modality to the auxiliary modality. Rather than manually designing fusion rules, Lei et al. (2023b) presented a multi-contrast VarNet ('MCVN') to explicitly model the relationship between different contrasts.

**Generative Models.** Generative models utilizing side information for MR image reconstruction are GAN-based and score-based algorithms. These models can be divided into reconstruction and synthesis methods, in which the former is our focus. Specifically, Dar et al. (2020) utilized conditional GANs with three priors—shared high-frequency, low-frequency, and perceptual priors. Kelkar & Anastasio (2021) proposed a framework for estimating objects from incomplete imaging measurements by optimizing in the latent space of a style-based generative model, using constraints from a related prior image. Levac et al. (2023) introduced a score-based generative model ('DMSI') to learn a joint Bayesian prior over multi-contrast data.

Despite significant advancements, existing methods for LIPs still struggle with highly under-sampled data, often leading to degraded image quality or hallucinations. The former can be attributed to the lack of efficiency in exploiting side information, while the latter represents over-reliance on it. These limitations highlight the need for a more principled approach that can efficiently integrate additional context, maintaining consistency with acquired data while minimizing artifacts.

## 4 TRUST GUIDED VARIATIONAL NETWORK (TGVN)

We now give details of our proposed method, that effectively and reliably leverages *side information* to impose additional constraints into the LIP and guide the solution to fall within a contextually-appropriate distribution. In this setting, we assume that we have access to the additional side information denoted by $\mathbf{s}$ when solving for $\mathbf{x}$ using the system of equations $\mathcal{A}\mathbf{x} = \mathbf{k}$. So long as $\mathbf{s}$ and $\mathbf{x}$ are conditionally dependent given $\mathbf{k}$ (i.e., the conditional mutual information $I(\mathbf{s}; \mathbf{x} \mid \mathbf{k}) > 0$), the knowledge of $\mathbf{s}$ can be exploited to reduce the uncertainty in estimating $\mathbf{x}$ from $\mathbf{k}$ (Cover & Thomas, 2006). As such, our solution assumes the existence of such conditional dependence.

### 4.1 THE MOTIVATION: AMBIGUOUS SPACE CONSISTENCY

Deep learning and physics-based unrolled networks have shown notable success in MR image reconstruction from sparse $k$-space data (Knoll et al., 2020a; Muckley et al., 2021), primarily due to their ability to enforce *data consistency*—ensuring that the reconstructed images closely match the acquired measurements. However, while data consistency is crucial for aligning the solution with the observed data, it might not be enough to resolve inherent ambiguities in the solution space, particularly at higher accelerations where an abrupt degradation in image quality has been highlighted (Radmanesh et al., 2022), rendering the images non-diagnostic. To address this issue, we introduce

the concept of *ambiguous space consistency*, which goes beyond data consistency and complements it. Essentially, our idea is to identify a subspace of images that could significantly alter reconstruction quality without substantially affecting the objective function of the MR image reconstruction problem.

Let $\mathbf{x}_p$ be a particular solution to the equation $\mathcal{A}\mathbf{x} = \mathbf{k}$ and $U\Sigma V^H$ represent the singular value decomposition (SVD) of $\mathcal{A}$, where $U$ and $V$ are unitary matrices, and $\Sigma$ is a rectangular diagonal matrix with singular values sorted in descending order along its diagonal. Given a small positive threshold $\delta$, we define the *ambiguous space* as the subspace spanned by the right singular vectors (columns of $V$) with corresponding singular values smaller than $\delta$ and denote it as $\mathcal{W}_\delta(\mathcal{A})$. Observe that if we add any unit vector $\mathbf{x}_a \in \mathcal{W}_\delta(\mathcal{A})$ to $\mathbf{x}_p$, the data consistency distance $\|\mathcal{A}(\mathbf{x}_p + \mathbf{x}_a) - \mathbf{k}\|_2^2$ can at most be $\delta^2$. In other words, perturbing a solution that aligns with the observed measurements by adding a vector from the ambiguous space results in only a minor change to the objective value. However, only certain $\mathbf{x}_a$ maximize the desired similarity between $\mathbf{x}_p + \mathbf{x}_a$ and $\mathbf{x}^*$, indicating that, once a particular solution is found, images from $\mathcal{W}_\delta(\mathcal{A})$ introduce ambiguity in the reconstruction problem. That is, they might visually alter the reconstruction quality without significantly affecting the loss. Inspired by this observation, we propose to explicitly learn a constraint that removes undesirable solutions from $\mathcal{W}_\delta(\mathcal{A})$. Our idea is to project $\mathbf{x}$ onto $\mathcal{W}_\delta(\mathcal{A})$ with the orthogonal projector $\mathcal{P}_\delta$ and to guide $\mathbf{x}$ to be maximally consistent with the side information $\mathbf{s}$ using a learnable module $\mathcal{H}$ parametrized by $\gamma$. Specifically, we add a squared Euclidean distance constraint $\|\mathcal{P}_\delta \mathbf{x} - \mathcal{H}(\mathbf{s}; \gamma)\|_2^2$ to equation 2 to obtain

$$\widehat{\mathbf{x}} = \arg\min_{\mathbf{x}} \frac{1}{2}\|\mathcal{A}\mathbf{x} - \mathbf{k}\|_2^2 + \frac{\beta}{2}\|\mathcal{P}_\delta \mathbf{x} - \mathcal{H}(\mathbf{s}; \gamma)\|_2^2 + \Psi(\mathbf{x}). \tag{4}$$

Our reason for choosing a more general projector $\mathcal{P}_\delta$ rather than simply using an orthogonal projector onto the null space of $\mathcal{A}$ is twofold. First, in practice, the matrix $\mathcal{A}$ (forward operator) in parallel MR imaging and other LIPs can have a trivial null space but still exhibit many small, non-zero singular values, thereby making the null space approach ineffective. This is the reason for high noise amplification at higher acceleration rates (Pruessmann et al., 1999). Second, even when the null space is non-trivial (i.e., it does not only contain the zero vector), the presence of small singular values can pose challenges, and the proposed approach can further assist in resolving these ambiguities.

## 4.2 THE SOLUTION: ITERATIVE OPTIMIZATION

Similar to E2E-VarNet, the solution to the equation 4 can be obtained iteratively by unrolling the network for a fixed number of times. As the added constraint involves only a squared Euclidean distance, its integration into equation 3 is straightforward. Starting with $\mathbf{x}_0 = \mathcal{A}^H \mathbf{k}$, we execute the following sequence of steps for a total of $T$ iterations.

$$\mathbf{x}^{t+1} = \mathbf{x}^t - \eta^t \mathcal{A}^H\left(\mathcal{A}\mathbf{x}^t - \mathbf{k}\right) \underbrace{-\mu^t \mathcal{P}_\delta\left(\mathbf{x}^t - \mathcal{H}(\mathbf{s}; \gamma^t)\right)}_{\text{trust-guidance}} - \Phi\left(\mathbf{x}^t; \theta^t\right). \tag{5}$$

At the end of the iteration $T$, we obtain $\mathbf{x}^T$ parameterized by $\Omega \triangleq \Theta \cup \{\delta, \gamma^0, \ldots, \gamma^{T-1}, \mu^0, \ldots, \mu^{T-1}\}$. Assuming access to ground truth $\mathbf{x}^*$, the parameters $\Omega$ are learned in a supervised manner to maximize a desired similarity between $\mathbf{x}^T$ and $\mathbf{x}^*$. It is worth noting that the parameter $\delta$ can be learned from the data as proposed, or it can be fixed based on the coil specifications and under-sampling pattern by analyzing the distribution of singular values.

In high-dimensional problems like parallel MR imaging, the computational burden of working directly with large-scale operators can be prohibitive. Therefore, instead of explicitly calculating the SVD of the forward operator, which would be computationally expensive, we seek an efficient alternative. Here, we present an efficient approximation of the exact orthogonal projector $\mathcal{P}_\delta$, which bypasses the need for SVD computation. This approach is crucial for managing the scale of the forward operator, which may contain hundreds of thousands of rows and columns, making explicit methods infeasible. For a set $\mathcal{K}$, Let $1_\mathcal{K}(x)$ denote an indicator function that equals 1 if $x \in \mathcal{K}$ and 0 otherwise. Given $\delta$, the exact projector can be written as $\mathcal{P}_\delta = \sum_i 1_{[0,\delta)}(\sigma_i)\mathbf{v}_i\mathbf{v}_i^H$. Instead of assigning binary weights to the $i$th projection, we can weigh them by $\delta^2/(\delta^2 + \sigma_i^2)$, and define

$$\mathcal{P}_\delta' \triangleq \sum_i \frac{\delta^2}{\delta^2 + \sigma_i^2}\mathbf{v}_i\mathbf{v}_i^H = \left(I + \frac{1}{\delta^2}\mathcal{A}^H\mathcal{A}\right)^{-1}, \tag{6}$$

which can be implemented using the Conjugate Gradient (CG) method (Hestenes et al., 1952).

## 5 EMPIRICAL VALIDATION

We validated the efficacy of TGVN by using it for multi-coil MR image reconstruction from different contrasts across different anatomies and field strengths. In all experiments, we utilized the efficient approximate projection introduced in equation 6. In our empirical validation, we seek answers to the following four questions: **(Q1)** Is there any benefit in using the side information when solving the task? **(Q2)** How effective is TGVN at utilizing the side information? **(Q3)** Does projecting onto the *ambiguous space* provide any benefits compared to no projection? **(Q4)** How robust is the proposed approach to different under-sampling factors, misregistration, and degradation? To answer Q1, we compare the reconstruction performance of TGVN against E2E-VarNet of the same capacity that do not utilize side information. Q2 is answered by comparing the performance of TGVN against several recent deep-learning baselines that also leverage side information in image reconstruction: MTrans (Feng et al., 2022), MCVN (Lei et al., 2023b), and DMSI (Levac et al., 2023). To address Q3, we compare the performance of TGVN with and without the projection. Q4 is answered by conducting experiments using multiple under-sampling factors and deliberately introduced misregistrations. We present our findings related to the first and second questions in Section 5.1.1 and 5.1.2, and our findings related to the third and fourth questions in Appendix A, which show that projection onto the ambiguous space significantly improves TGVN's performance and that TGVN is robust to moderate under-sampling and misregistration of side information.

In our experiments, under-sampling was implemented along the phase-encoding direction. The target images were selected as the root-sum-of-squares (RSS) combination $\sqrt{\sum_i |\mathbf{x}_i|^2}$ of fully-sampled component coil images $\mathbf{x}_i$. We evaluated the reconstruction quality using three metrics: the structural similarity index (Wang et al., 2004) (SSIM), peak signal-to-noise ratio (PSNR), and normalized root-mean-squared error (NRMSE). For the SSIM metric, a $7 \times 7$ uniform kernel was utilized, along with the standard $k-$values of $0.01$ and $0.03$. The range parameter is given as input to the SSIM calculation and is set to the maximum pixel value of the corresponding volume. To demonstrate the statistical significance of the improvements in image reconstruction metrics, we performed a Wilcoxon signed-rank test (Wilcoxon, 1945) between the metrics calculated on the test dataset for TGVN ($s_{\text{ours}}$) and the best-performing baseline[1] ($s_{\text{base}}$). Let $\mathcal{D}$ be the distribution of pair-wise difference $s_{\text{ours}} - s_{\text{base}}$. Then under the alternate hypothesis, $\mathcal{D}$ is "stochastically greater than a distribution symmetric about zero" for SSIM and PSNR and "stochastically less than a distribution symmetric about zero" for NRMSE. Additional training and evaluation details are presented in Appendix B.

### 5.1 DATASETS

In knee experiments, we utilized a subset of the multi-coil track of the fastMRI knee dataset: an open-source dataset consisting of $k$-space measurements from clinical 3T and 1.5T scanners paired with the ground truth clinical cross-sectional images (Zbontar et al., 2018). The dataset comprises coronal MR scans of $428$ patients using the proton-density weighting with fat suppression (PDFS) and proton density without fat suppression (PD). Data acquisition employed a 15-channel knee coil array in conjunction with a standard Cartesian 2D Turbo Spin Echo (TSE) protocol, routinely used at the provider institution of fastMRI (Knoll et al., 2020b). The dataset comprises of $368$, $30$, and $30$ volumes for training, validation, and test, respectively, with a total of $15,231$ slices.

Brain experiments utilized the M4Raw dataset (Lyu et al., 2023): a publicly available multi-channel $k$-space dataset of brain scans of $183$ healthy volunteers acquired using a low-field ($0.3$T) scanner. It includes axial MR scans with three contrasts, acquired using a $4$-channel array: T1-weighted (T1w), T2-weighted (T2w), and fluid-attenuated inversion recovery (FLAIR). Each scan has $18$ slices per contrast with varying repetitions. We used measurements from single repetition to reconstruct multi-repetition aggregated RSS targets. The training, validation, and test sets have $128$, $30$, and $25$ volumes.

---

[1] $s_{\text{base}}$ is using the scores of the baseline with the best average score.

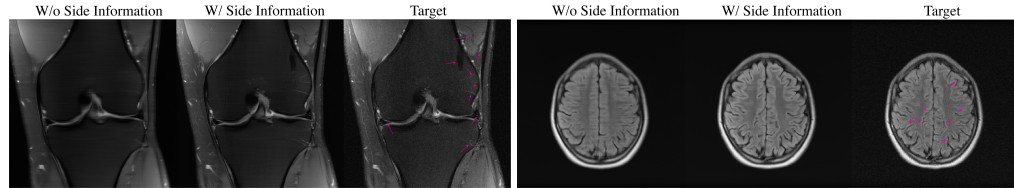

| W/o Side Information | W/ Side Information | Target | W/o Side Information | W/ Side Information | Target |

(a) Coronal PDFS image reconstruction without and with side information at $20\times$ acceleration

(b) Axial FLAIR image reconstruction without and with side information at $9\times$ acceleration

Figure 3: **Leveraging side information significantly enhances the reconstruction quality. Left:** Reconstructed MR image from highly sparse MR measurements using E2E-VarNet. **Middle:** Reconstructed MR image from the same sparse MR measurements, with additional side information from a different sequence using TGVN (having the same capacity as the E2E-VarNet). **Right:** Ground truth target image, with prominent anatomical features highlighted by purple arrows.

### 5.1.1 KNEE EXPERIMENTS

In our experiments involving knee MR images, we treat the highly under-sampled PDFS-weighted $k$-space measurements as the "main information" and reconstruct a PDFS-weighted RSS image from them, using the corresponding moderately under-sampled PD $k$-space measurements (which we treat as "side information"). To evaluate TGVN's effectiveness in diverse settings, we conducted two experiments with different sampling rates in main and side information, featuring both the non-trivial and trivial null space cases, respectively. The detailed results are provided in Appendix C.1.

**Set I: $20\times$ Under-sampled Main Information and $2\times$ Under-sampled Side Information.** We applied a $20\times$ under-sampling with a $3\%$ fully-sampled center random mask to the PDFS measurements, and a $2\times$ equispaced under-sampling with no fully-sampled center to the PD measurements. Fig. 3a shows the reconstruction results for coronal PDFS images with and without using the side-information. At $20\times$ acceleration, side information aids the reconstruction significantly while reconstruction without it loses various essential features. Fig. 4 shows the reconstructions from TGVN and multiple baselines that use side information. MTrans and MCVN exhibit significant blurring of anatomical features, and DMSI suffers severely from noise amplification, which is seen clearly in the absolute difference images. In contrast, the output of TGVN is significantly superior: both the sharpness and the details are better preserved in the TGVN approach. Furthermore, the meniscus tear region is distinctly more noticeable with TGVN, highlighting that it is *more effective* in leveraging the side information to preserve key features in the image despite highly-sparse measurements.

**Set II: $14\times$ Under-sampled Main Information and $3\times$ Under-sampled Side Information.** We applied a $14\times$ under-sampling with a $3\%$ fully-sampled center random mask to the PDFS measurements. Additionally, a $3\times$ equispaced under-sampling with no fully-sampled center was applied to the PD measurements. It is worth noting that knee images are acquired with 15 coils, which implies that, in this experiment, the null space is trivial, i.e., it contains only the zero vector. Therefore, methods utilizing the range-null space decomposition are unlikely to be effective. Fig. 2 illustrates example input, side information, and target images for this experiment.

Table 1 reports the quantitative evaluation of TGVN against the baselines in both sets of experiments. TGVN achieves the best average score across all metrics, proving that it is more effective than baselines utilizing side information. In each experiment and for each evaluation metric SSIM, PSNR, NRMSE; Wilcoxon signed-rank test rejected the null hypothesis at a confidence level of $5\%$, concluding that there is a statistically significant difference between $s_{\text{ours}}$ and $s_{\text{base}}$. We provide additional details regarding the evaluation results and reconstructions in Appendix C.1 and D, showing that TGVN has superior performance for *almost all examples* in the test dataset for each experiment.

### 5.1.2 BRAIN EXPERIMENTS

In our experiments using brain MR images, we use the highly under-sampled FLAIR $k$-space measurements from a single repetition as the "main information" and aim to reconstruct multi-repetition-averaged[2] FLAIR RSS images. As "side information" we use the corresponding low-SNR, single-

---

[2] multiple acquisitions of the same anatomy, which are averaged to improve image quality by reducing noise

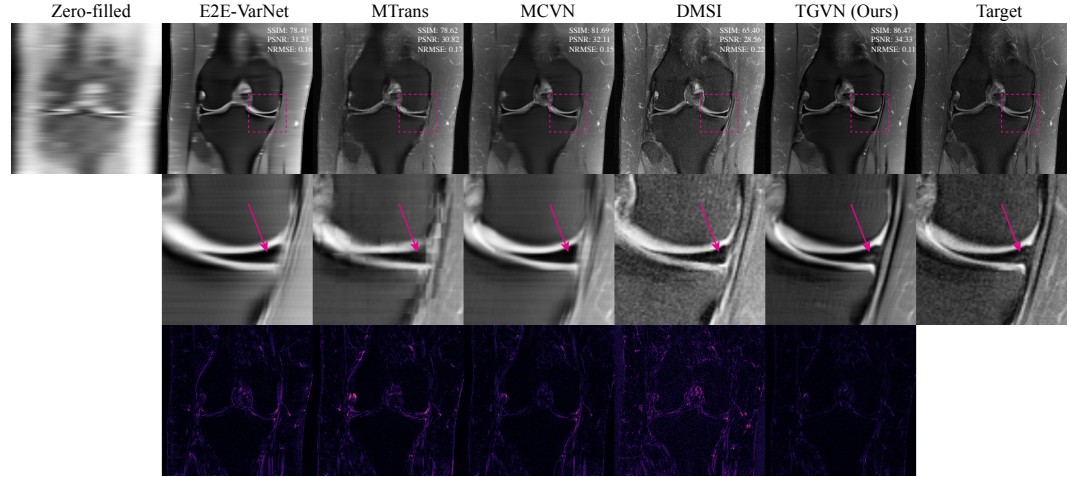

Figure 4: Reconstructions from Set I showing the effectiveness of TGVN in leveraging side information. TGVN is able to reconstruct a high-quality image even at challenging acceleration levels of $20\times$, in comparison to other baselines. The *meniscus tear*, illustrated in the ground truth image and reconstructions with a purple arrow, is clearly visible in TGVN reconstruction. Top row: Original reconstructions of various methods. Middle row: Zoomed-in regions from the upper right corner of the images for better visualization. Bottom row: Absolute differences between each reconstruction and the ground truth, with a consistent color mapping to highlight error magnitudes.

repetition T2-weighted (T2w) *k*-space measurements. We selected FLAIR as the main information and T2w as the side information because, in the protocol described by Lyu et al. (2023), FLAIR has the longest acquisition time per repetition (135 seconds), while T2w has the shortest (71.5 seconds). Note that the protocol includes two repetitions for FLAIR and three for T2w. Hence, by using a single repetition as side information, we achieve a practical acceleration factor of $3\times$.

**Set III: $18\times$ Under-sampled Main Information and Single-repetition Side Information.** We applied a $18\times$ under-sampling with a $2\%$ fully-sampled center equispaced mask[3] to the FLAIR measurements from single repetition, achieving a practical acceleration factor of $36\times$. We chose the equispaced mask to evaluate the proposed method in a more diverse setting, as random masks were used in the knee experiments. T2w images are used as fully-sampled due to the low SNR and small matrix size of the acquisition (Lyu et al., 2023). Fig. 3b shows the reconstruction results for axial FLAIR images with and without using the side information. At $9\times$ acceleration, side information aids the reconstruction significantly while reconstruction without it loses various essential features. Fig. 5 demonstrates the reconstruction results for axial FLAIR images. Side information enables achieving decent image quality at the challenging acceleration level of $18\times$. TGVN demonstrates superior performance in integrating this information compared to other methods, as evidenced by the enhanced depiction of anatomical features in the zoomed-in region and the consistently better reconstruction metrics. Furthermore, for each evaluation metric SSIM, PSNR, NRMSE; Wilcoxon signed-rank test rejected the null hypothesis at a confidence level of $5\%$, concluding that there is a statistically significant difference between $s_{\text{ours}}$ and $s_{\text{base}}$. Table 1 reports the quantitative evaluation results of TGVN and the baselines, in which TGVN achieves the best average score across all metrics. The statistically significant performance difference between TGVN and the other baselines indicates the side information is beneficial in guiding the reconstruction, and TGVN is more effective at leveraging it. The detailed evaluation results and reconstructions are provided in Appendix C.2 and D, demonstrating statistically significant improvements.

*Remark:* We omitted the DMSI baseline to avoid biased comparisons, as DMSI generates multi-repetition-averaged images only when trained on such inputs. These experiments, however, reconstruct multi-repetition-averaged images from single-repetition inputs, accelerating both the main and side information. Training DMSI on single-repetition inputs would favor TGVN and other baselines, while training it on multi-repetition-averaged inputs would give DMSI an unfair advantage.

---

[3]To achieve $18\times$ overall acceleration, the spacing between adjacent outer samples is set to 32.

Table 1: **Quantitative evaluation results in terms of SSIM, PSNR, and NRMSE for the knee and brain experiments:** For each method, the mean and standard error of the mean over the test dataset are reported. Bold statistics indicate the best performance in each category. For SSIM and PSNR, higher is better; for NRMSE, lower is better.

| | Exp. | TGVN | DMSI | MCVN | MTrans | E2E-VarNet |
|---|---|---|---|---|---|---|
| **SSIM** | I | **84.92 ± 0.19** | 56.99 ± 0.31 | 82.89 ± 0.21 | 80.84 ± 0.23 | 81.33 ± 0.23 |
| | II | **85.52 ± 0.19** | 58.76 ± 0.31 | 83.13 ± 0.21 | 81.25 ± 0.22 | 83.40 ± 0.21 |
| | III | **87.34 ± 0.12** | | 86.95 ± 0.12 | 84.03 ± 0.14 | 75.63 ± 0.18 |
| **PSNR** | I | **30.92 ± 0.07** | 22.22 ± 0.10 | 29.97 ± 0.07 | 28.93 ± 0.07 | 29.30 ± 0.07 |
| | II | **31.31 ± 0.07** | 22.68 ± 0.10 | 30.07 ± 0.07 | 29.11 ± 0.07 | 30.37 ± 0.07 |
| | III | **30.81 ± 0.08** | | 30.75 ± 0.08 | 28.70 ± 0.08 | 24.60 ± 0.09 |
| **NRMSE** | I | **0.14 ± 0.001** | 0.40 ± 0.004 | 0.16 ± 0.001 | 0.18 ± 0.001 | 0.17 ± 0.001 |
| | II | **0.13 ± 0.001** | 0.38 ± 0.004 | 0.16 ± 0.001 | 0.17 ± 0.001 | 0.15 ± 0.001 |
| | III | **0.158 ± 0.002** | | 0.159 ± 0.002 | 0.201 ± 0.002 | 0.32 ± 0.004 |

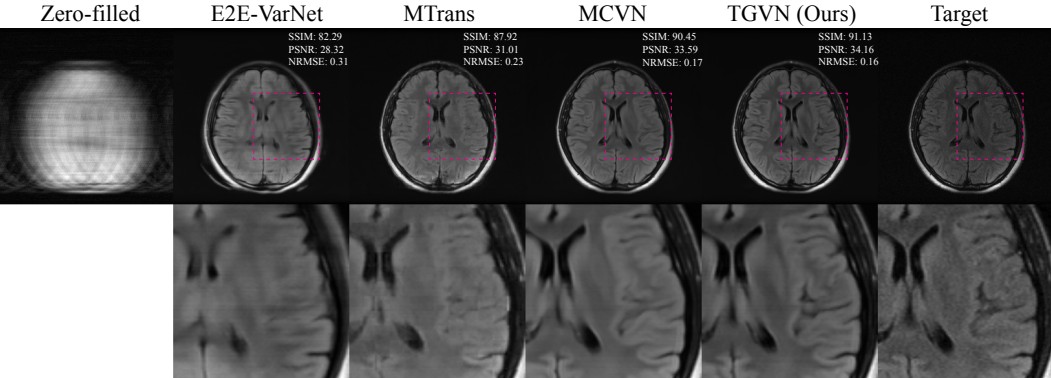

Figure 5: Reconstructions from Set III showing the effectiveness of TGVN at challenging acceleration level of $18\times$ when reconstructing brain images, in comparison to the baselines. Top row: Original reconstructions of zero-filled, E2E-VarNet, MTrans, MCVN, and TGVN methods, followed by the ground truth image. Bottom row: Zoomed-in regions from the upper right corner of each reconstruction and the ground truth, upscaled for better visualization.

## 6 CONCLUSION

Our work introduces a novel framework, the **T**rust-**G**uided **V**ariational **N**etwork (**TGVN**), that demonstrates the power of leveraging side information in solving LIPs, with specific application to the MR image reconstruction problem. By learning to eliminate solutions from the *ambiguous space* and remaining faithful to the acquired measurements through *data consistency*, our principled approach maximally utilizes the side information and attempts to minimize the risk of hallucinations. Our key finding is that, when incorporated effectively, side information can significantly improve reconstruction quality and preserve key anatomical and pathological features, even at exceedingly high under-sampling regimes. These findings can have a transformative impact in healthcare by enabling widespread access to MR imaging for diagnosing diseases at the population level.

**Limitations and Future Work.** While the results for MR image reconstruction using TGVN are very promising, there are various limitations of the current work, which will inform our future research directions. First, in our experiments, we only used complementary-contrast measurements from the same MR examination as side information. In the future, we intend to explore incorporation of different types of side information, including a patient's prior scans and associated textual data (e.g., clinical notes and medical history), as well as features learned from related tasks.

**Reproducibility Statement.** We have made significant efforts to ensure the reproducibility of our work. The code and scripts for model training and evaluation have been prepared and will be made publicly available in a repository after the review phase. We used the fastMRI and M4Raw public datasets in our experiments; instructions for accessing these datasets are detailed in (Zbontar et al., 2018) and (Lyu et al., 2023), respectively. Hyperparameters and configurations are provided in Appendix B and will also be included in the repository. Pretrained model checkpoints will be available upon request.

**Ethics Statement.** We acknowledge and adhere to the ICLR Code of Ethics in all aspects of this submission, including research design, data usage, and reporting. Our study utilizes publicly available datasets (fastMRI, M4Raw) with due consideration of their ethical guidelines and without involving any protected health information. We have taken care to address issues related to potential biases, fairness, and the responsible application of our methods. Our research is intended solely for scientific purposes within the field of medical imaging and does not present immediate risks of misuse or inappropriate application. We remain committed to responsible research practices and encourage the community to engage with our work in an ethical and transparent manner.

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

# A  ABLATION STUDIES

In the ablation studies, we try to answer the following two questions: **(Q3)** Is there any benefit in using the projection onto the *ambiguous* space? **(Q4)** How robust is TGVN with degraded side information such as misregistation or under-sampling? To answer the first question, we compare our proposed method with and without the projection in Sec. A.1, i.e., we compare the reconstruction performance of the unrolled network implementing equation 5 and the unrolled network implementing a modified version of equation 5 in which $\mathcal{P}_\delta$ is replaced by the identity operator. In other words, the network without the projection implements the following update equations for $T$ iterations, starting with $\mathbf{x}_0 = \mathcal{A}^H \mathbf{k}$.

$$\mathbf{x}^{t+1} = \mathbf{x}^t - \eta^t \mathcal{A}^H \left( \mathcal{A} \mathbf{x}^t - \mathbf{k} \right) - \mu^t \left( \mathbf{x}^t - \mathcal{H}(\mathbf{s}; \gamma^t) \right) - \Phi \left( \mathbf{x}^t; \theta^t \right). \tag{7}$$

The second question is answered by conducting two experiments. In the misregistration ablation study (Sec. A.2.1), we compare the performance of TGVN when the side information is perfectly registered versus random misregistrations simulated by small random shifts and rotations during training and/or inference time. In the under-sampling ablation study (Sec. A.2.2), we compare the performance of three models: (I) TGVN utilizing under-sampled side information, (II) TGVN utilizing under-sampled side information reconstructed first with an E2E-VarNet, and (III) TGVN utilizing fully sampled side information. We note that the TGVNs in (I), (II), and (III) have the same number of parameters; however, the inclusion of the E2E-VarNet in (II) introduces a minor element of unfairness in the comparison.

## A.1  EFFECT OF PROJECTION

We applied a $9\times$ under-sampling with a $4\%$ fully-sampled center equispaced mask to the FLAIR measurements from single repetition, achieving a practical acceleration factor of $18\times$. That is, to achieve $9\times$ acceleration, the spacing between adjacent samples is set to $15$. As in experiment III, T2w images are used as fully-sampled due to the low SNR and small matrix size of the acquisition (Lyu et al., 2023). We performed three Wilcoxon signed-rank tests, and they rejected the null hypotheses at a confidence level of $5\%$, concluding that there is a statistically significant difference between $s_{\text{w/}}$ and $s_{\text{w/o}}$, where $s_{\text{w/}}$ and $s_{\text{w/o}}$ represent the SSIM, PSNR, and NRMSE scores calculated on the test dataset for the TGVN with and without the proposed projector, respectively. Fig. 6 presents the quantitative evaluation results for the effect of projection, demonstrating that the projection improves reconstruction quality for almost all slices in the test dataset.

## A.2  ROBUSTNESS TO DEGRADED SIDE INFORMATION

### A.2.1  MISREGISTRATION.

Similar to A.1, we applied a $9\times$ under-sampling with a $4\%$ fully-sampled center equispaced mask to the FLAIR measurements from single repetition, achieving a practical acceleration factor of $18\times$. At each slice, three random variables $dx$, $dy$, and $d\theta$ were drawn uniformly from the interval $[-4, 4]$, and side information is translated by $dx$ and $dy$ pixels and rotated by $d\theta$ degrees. As expected, we observed that if TGVN does not encounter misregistration during training, the reconstruction quality degrades sharply during inference. However, random and small misregistrations during training make it robust to small misregistrations during inference, as seen in Fig. 7, and it still achieves much better scores than E2E-VarNet of the same capacity. This observation is supported by Wilcoxon tests at a $5\%$ confidence level for each metric—SSIM, PSNR, and NRMSE—demonstrating a statistically significant performance difference in favor of TGVN encountering misregistered side information, compared to E2E-VarNet, which does not utilize side information. This observation is also supported by Fig. 8.

### A.2.2  UNDER-SAMPLING.

We applied a $14\times$ under-sampling with a $3\%$ fully-sampled center random mask to the PDFS measurements. Additionally, a $3\times$ equispaced under-sampling with no fully-sampled center was applied to the PD measurements in one TGVN (I), while in another, the side information was given as fully-sampled (III). In addition, (II) employs a two-stage reconstruction, where the $3\times$ under-sampled side information is first reconstructed with an E2E-VarNet with 30 million trainable parameters.

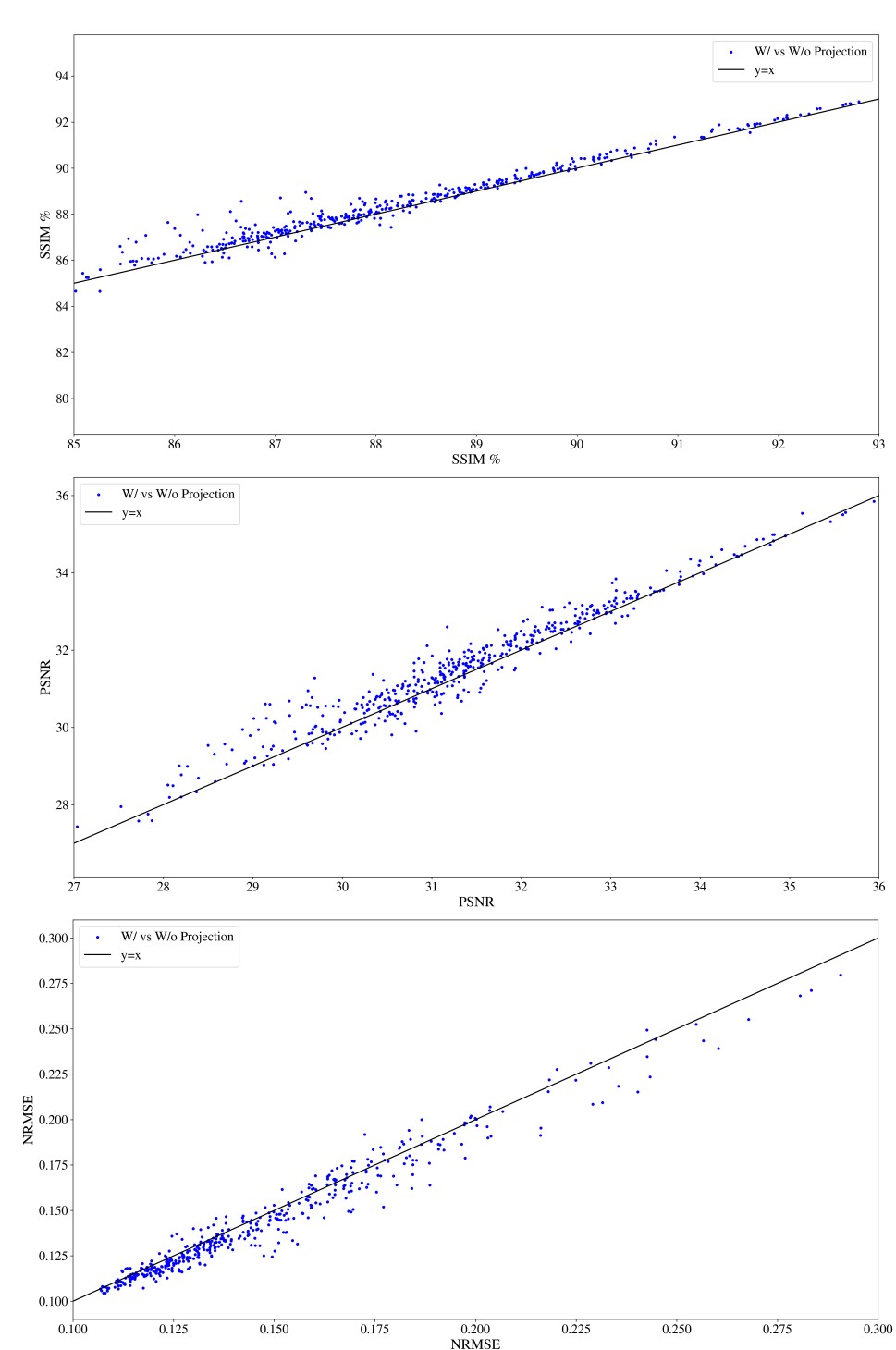

Figure 6: **Quantitative evaluation results in terms of SSIM %, PSNR, and NRMSE over the test dataset for the Ablation Study A.1**. Each blue point has x- and y-coordinates representing values achieved by TGVN without and with the proposed projector, respectively. The ideal scenario is that for all samples in the test dataset, the proposed projector leads to better scores. That is, the blue points are always above the $y = x$ line for SSIM and PSNR, and always below the $y = x$ line for NRMSE. TGVN achieves **better** performance for **almost all slices** in the test dataset.

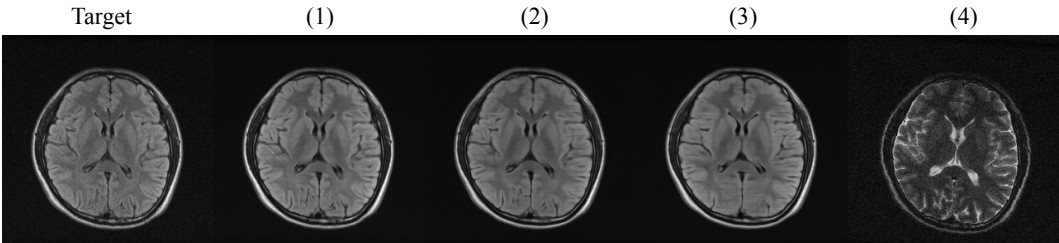

Figure 7: **Ablation Study A.2.1**. (1): Reconstruction from TGVN trained with registered side information, encountering registered side information during inference. (2): Reconstruction from TGVN trained with augmentations simulating misregistrations, encountering misregistered side information (4) during inference. (3): Reconstruction from E2E-VarNet without access to side information. (4): Misregistered side information during inference. **Despite randomly misregistered side information, (2) preserves anatomical details much better than (3).**

Table 2: **Quantitative evaluation results for the Ablation Study A.2.2 in terms of SSIM, PSNR, and NRMSE:** The mean and standard error of the mean over the test dataset are reported. For SSIM and PSNR, higher is better; for NRMSE, lower is better. Bold statistics indicate the method achieving the best performance for each metric.

| Metric/Model | TGVN with 3x under-sampled side information | TGVN with 3x under-sampled side information, the two-stage approach | TGVN with fully-sampled side information |
|:---:|:---:|:---:|:---:|
| **SSIM** | $85.52 \pm 0.19$ | $85.56 \pm 0.19$ | $\mathbf{85.97 \pm 0.18}$ |
| **PSNR** | $31.31 \pm 0.07$ | $31.41 \pm 0.07$ | $\mathbf{31.60 \pm 0.07}$ |
| **NRMSE** | $0.13 \pm 0.001$ | $0.13 \pm 0.001$ | $0.13 \pm 0.001$ |

We observed that the reconstruction scores improve with fully-sampled side information, though the improvements are not easily noticeable. Furthermore, despite having 30 million more trainable parameters, the two-stage approach did not provide statistically significant improvements compared to TGVN utilizing under-sampled side information. The quantitative evaluation results for this experiment are provided in Table 2, and an example reconstruction from this study is shown in Fig. 9. Our takeaway from this experiment is that while fully-sampled side information provides the greatest benefit, moderately under-sampled side information is still helpful and significantly improves the reconstruction compared to not having any side information, cf. Table 1, E2E-VarNet column. Furthermore, an end-to-end training with under-sampled side information performs as good as the two-stage approach—reconstructing first the under-sampled side information, and using the reconstructed side information in TGVN.

## B  TRAINING AND EVALUATION DETAILS

### B.1  TGVN.

We optimized the MS-SSIM-L1 (Zhao et al., 2017; Wang et al., 2003) loss function using the ADAM optimizer (Kingma & Ba, 2014), with batch size of one per GPU and with default parameters, that employs a uniform kernel of size $33 \times 33$ and $k-$values of $0.01$ and $0.03$, across 5 values of $\sigma$ $(0.5, 1.0, 2.0, 4.0, 8.0)$, in both training and validation phases. The loss is calculated between the reconstructed and the ground truth root-sum-of-squares (RSS) Larsson et al. (2003) images. A starting learning rate of $3 \times 10^{-4}$ was used with exponential decay with parameters $0.98$. These parameters were determined through a grid search on the validation set. The training spanned 100 epochs, with the best model parameters selected based on the loss on the validation set. All the models were trained and tested on $4\times$ NVIDIA A100 GPUs using PyTorch for 10 days in knee, 2 days in brain experiments each with a unit batch size per GPU.

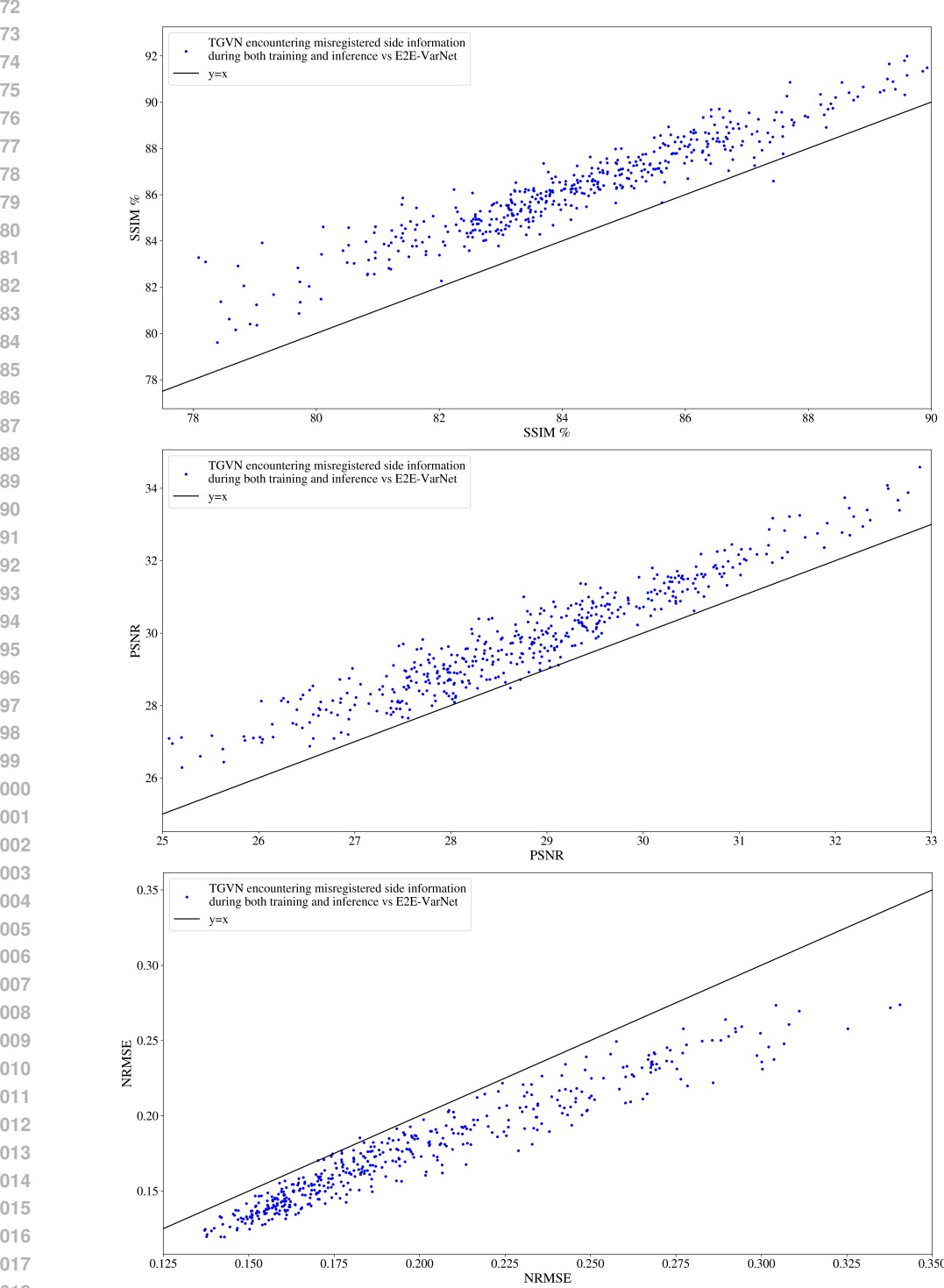

Figure 8: **Quantitative evaluation results in terms of SSIM %, PSNR, and NRMSE over the test dataset for the Ablation Study A.2.1**. Each blue point has x- and y-coordinates representing values achieved by E2E-VarNet without side information and TGVN encountering randomly misregistered side information during training and inference, respectively. The ideal scenario is that for all samples in the test dataset, TGVN leads to better scores despite **randomly misregistered side information**. That is, the blue points are always above the $y = x$ line for SSIM and PSNR, and always below the $y = x$ line for NRMSE. TGVN achieves **better** performance for **almost all slices** in the test dataset.

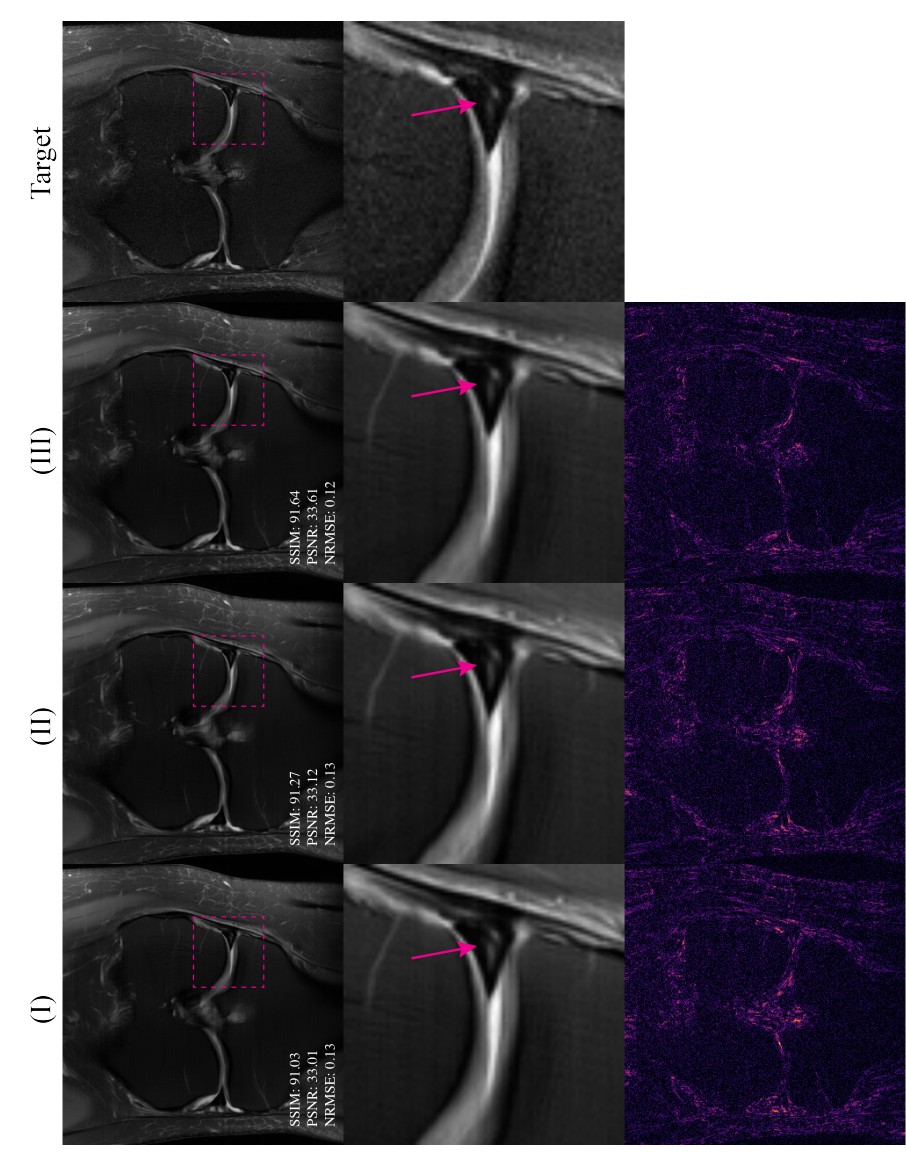

Figure 9: **Ablation Study A.2.2.** (I): Reconstruction from TGVN trained with 3× under-sampled side information. (II): Reconstruction from TGVN with 3× under-sampled side information, which is first reconstructed with an E2E-VarNet. (III): Reconstruction from TGVN trained with fully-sampled side information. While the performance metrics differ slightly, the *meniscus tear*, illustrated with purple arrows are visible in all reconstructions.

Both the refinement and SME modules are implemented using a U-net model (Ronneberger et al., 2015), which consists of four down-sampling and up-sampling paths, complemented by skip connections. Each path is equipped with two $3 \times 3$ convolutions, followed by instance normalization (Ulyanov et al., 2016), and leaky ReLU (Maas et al., 2013) activation functions. The first convolution layer outputs 21 channels in the refinement network and 8 in the SME network, with the number doubling in each subsequent layer, as in Sriram et al. (2021). In the brain experiments, the proposed TGVN comprises $T = 10$ TGVN blocks having approximately 67.3 million trainable parameters in total. In the knee experiments, it includes $T = 14$ TGVN blocks, resulting in about 94 million trainable parameters.

For enhanced numerical stability in training, complex-valued layer normalization and its inverse operator are used, similar to Sriram et al. (2021). Specifically, each network pass, denoted as $\mathbf{x} \mapsto \Phi(\mathbf{x})$, is executed as $\mathbf{x} \mapsto \mathcal{T}^{-1}(\Phi(\mathcal{T}\mathbf{x}))$. For an input tensor of shape (B, 2, H, W), where $B$ is the batch size, the two channels correspond to the real and imaginary components of the image. The normalization process $\mathcal{T}$ adjusts each sample so that the mean of each channel across the spatial dimensions $(H, W)$ is zero, and the standard deviation is set to one. Furthermore, $\mathcal{T}$ ensures that the real and imaginary channels are decorrelated, resulting in zero covariance between them. This is achieved by computing the $2 \times 2$ covariance matrix of the two channels and performing a linear combination with the transpose of the Cholesky decomposition of the inverse of the covariance matrix. This normalization step allows the network to handle the real and imaginary parts without any inherent bias or unintended correlation, ultimately improving the stability and performance of the model.

In both the knee and brain experiments, we implemented the approximate projector as described in Section 4.2 for the trust-guidance term due to computational constraints and learned the threshold parameter $\delta$ during training. For example, in Experiment Set II, the forward operator has more than $200,000$ rows and $200,000$ columns, making explicit SVD calculation prohibitively costly.

## B.2 BASELINES

For the baselines, we used the officially released repositories instead of re-implementing the models. As a result, we only needed to adjust the model capacities to match that of TGVN in each setting and modify the training learning rates to adapt to the fastMRI and M4Raw datasets.

### B.2.1 MTRANS

We used the CrossCMMT model and set the hidden dimension to 17 and 14 for the knee and brain experiments, respectively. Input sizes are chosen as the image matrix size, without any resolution change. Parameters `P1`, `P2` are set to 8, and `CTDEPTH` and `TRANSFORMER_NUM_HEADS` are set to 4, and `TRANSFORMER_MLP_RATIO` is set to 3. With these parameters, the knee model has 98.3 and the brain model has 66.1 million trainable parameters. Initial learning rate is set to $2 \times 10^{-4}$, and trained one model without scheduling and one model scheduled with an exponential decay $\gamma = 0.99$. All the models are trained for 100 epochs on $4\times$ NVIDIA A100 GPUs using PyTorch with a unit batch size per GPU. The best method is chosen according to the average validation SSIM. Training spanned approximately 7 days for knee and 1 day for brain.

### B.2.2 MCVN

We set the `in_channel` and `channel_fea` parameters to 264 and 224 without changing the default `iter_num` (4), which result in 94.1 and 67.8 million trainable parameters for the knee and brain experiments, respectively. For the brain experiments, we used an initial learning rate of $10^{-4}$, and trained one model without scheduling and one model scheduled with an exponential parameter 0.99. For the knee experiments, learning rates in the order of $10^{-4}$ resulted in unstable training, so we chose an initial learning rate of $10^{-5}$ and trained one model with the same scheduling and one model with the constant learning rate. Each model is trained for 100 epochs on $4\times$ NVIDIA A100 GPUs using PyTorch with a unit batch size per GPU and the best method is chosen according to the average validation SSIM. Training spanned approximately 7 days for knee and 1 day for brain.

### B.2.3 DMSI

By design, DMSI reconstructs complex-valued coil-combined images. From the reconstruction $\widehat{\mathbf{x}}_{\text{DMSI}}$, we obtained the reconstructed RSS combination using $\sum_i |S_i \widehat{\mathbf{x}}_{\text{DMSI}}|^2$ and compared against the ground-truth RSS images. To train the score network, we used SongUnet network architecture with positional embedding, and standard encoder and decoder. We did not modify the default parameters in the codebase, but only changed the `model_channels` to 210 to match the trainable number of parameters, resulting in 92.4 million parameters approximately. We employed augmentation with $\text{p} = 0.12$ and dropout with $\text{p} = 0.13$. We trained the model to minimize the EDM loss for 400,000 steps with a batch size of 4 per GPU. The training spanned approximately 10 days.

## C    DETAILED EVALUATION RESULTS

In this section, we present the quantitative evaluation results, calculated on the test dataset, for TGVN and the second-best method for each of Experiments I, II, and III as scatter plots, demonstrating TGVN's effectiveness over other baselines. Specifically, Fig. 10, Fig. 11, and Fig. 12 show the results for Experiments I, II, and III, respectively.

## C.1 KNEE EXPERIMENTS

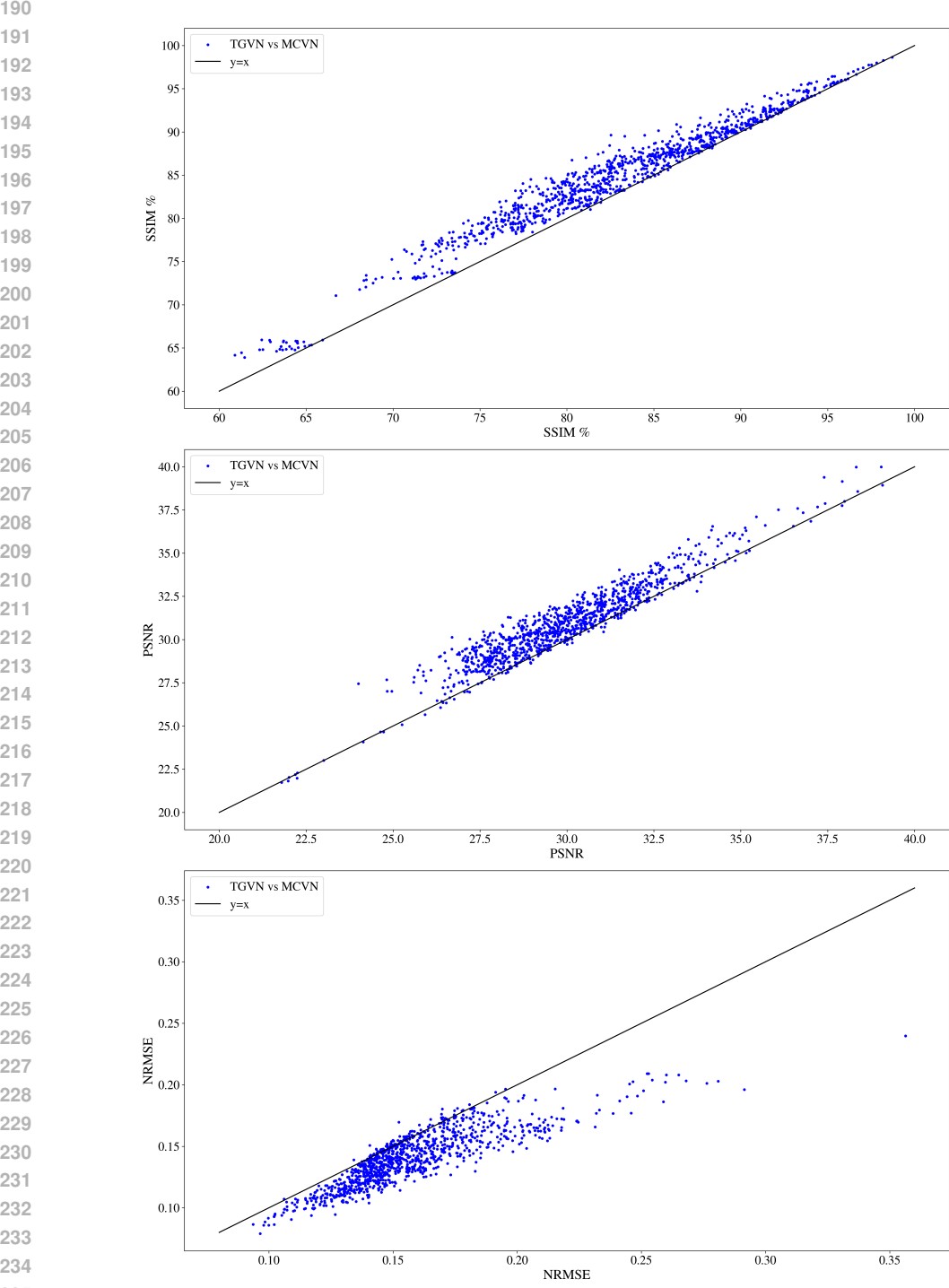

Figure 10: **Quantitative evaluation results in terms of SSIM %, PSNR, and NRMSE over the test dataset for Set I**. Each blue point has x- and y-coordinates representing values achieved by the second best performing method—MCVN and TGVN, respectively. The ideal scenario is that for all samples in the test dataset, the proposed method achieves better scores. That is, the blue points are always above the $y = x$ line for SSIM and PSNR, and always below the $y = x$ line for NRMSE. TGVN achieves **better** performance for **almost all slices** in the test dataset.

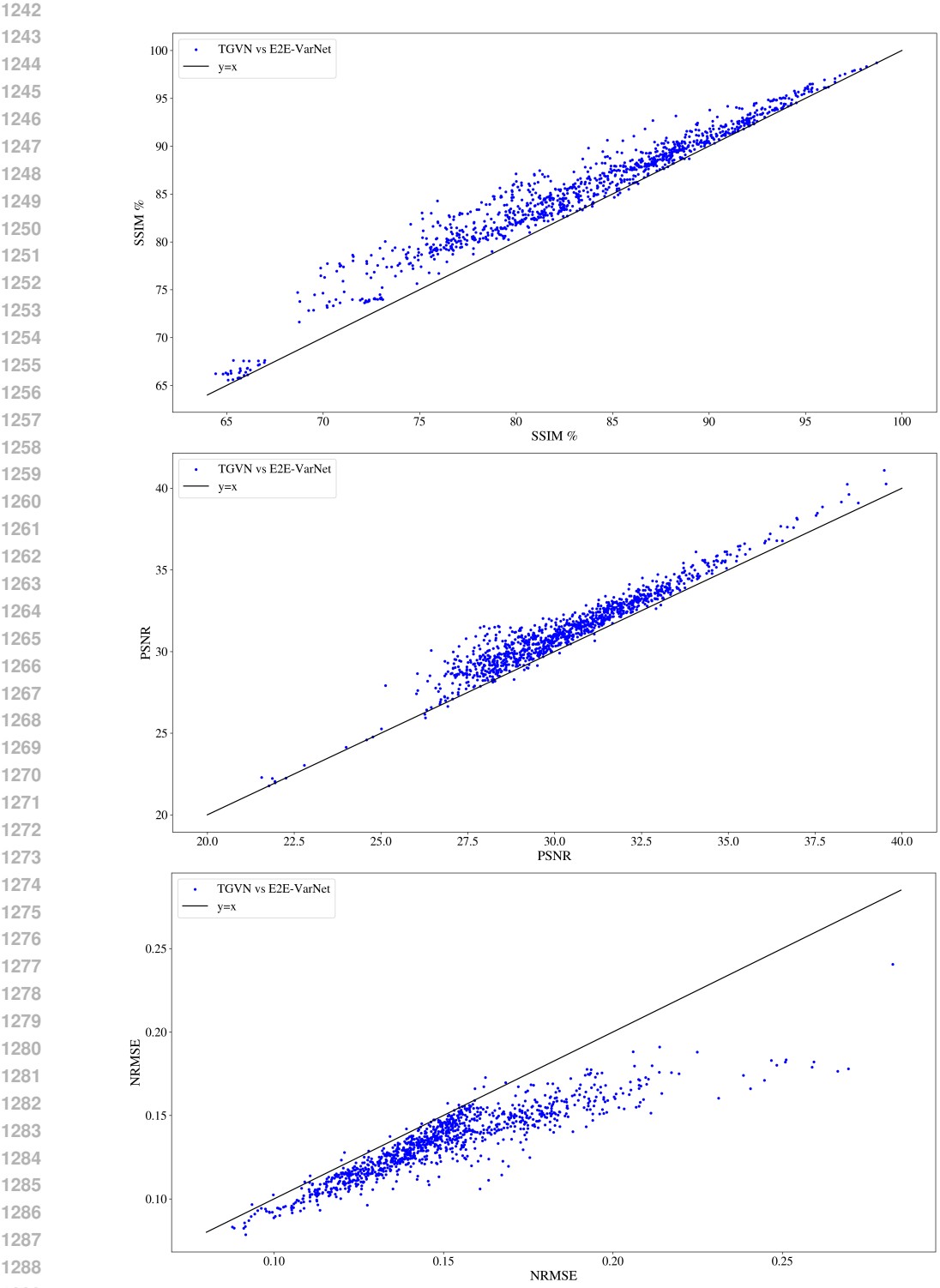

Figure 11: **Quantitative evaluation results in terms of SSIM %, PSNR, and NRMSE over the test dataset for Set II**. Each blue point has x- and y-coordinates representing values achieved by the second best performing method—E2E-VarNet and TGVN, respectively. The ideal scenario is that for all samples in the test dataset, the proposed method achieves better scores. That is, the blue points are always above the $y = x$ line for SSIM and PSNR, and always below the $y = x$ line for NRMSE. TGVN achieves **better** performance for **almost all slices** in the test dataset.

## C.2 Brain Experiments

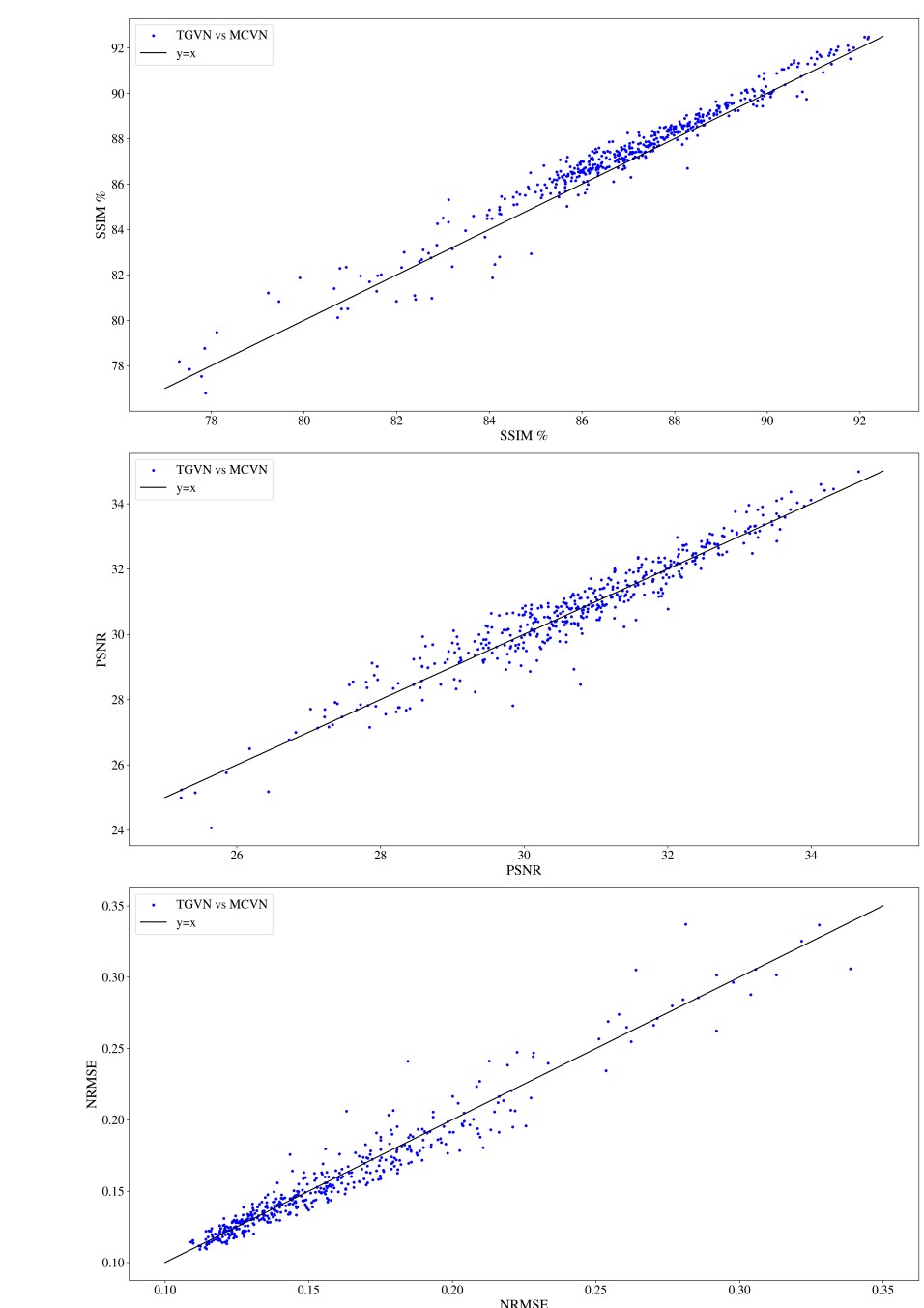

Figure 12: **Quantitative evaluation results in terms of SSIM %, PSNR, and NRMSE over the test dataset for Set III**. Each blue point has x- and y-coordinates representing values achieved by the second best performing method MCVN and TGVN, respectively. The ideal scenario is that for all samples in the test dataset, the proposed method achieves better scores. That is, the blue points are always above the $y = x$ line for SSIM and PSNR, and always below the $y = x$ line for NRMSE. In this case, the PSNR and NRMSE plots are visually closer than the SSIM plots. However, Wilcoxon signed-rank tests still reject the null hypothesis with $p$-values less than $7 \times 10^{-4}$ in each case, **demonstrating statistically significant improvements** with **TGVN**.

# D    FURTHER RECONSTRUCTION EXAMPLES

In this section, we provide the full-size versions of the reconstructions shown in the main text. Specifically, the reconstructions in Fig. 3, Fig. 4, and Fig. 5 are presented as their corresponding larger versions in Fig. 13, Fig. 14, and Fig. 17, respectively. Additionally, we include an example that was not provided in the main text, which is shown in Fig. 15.

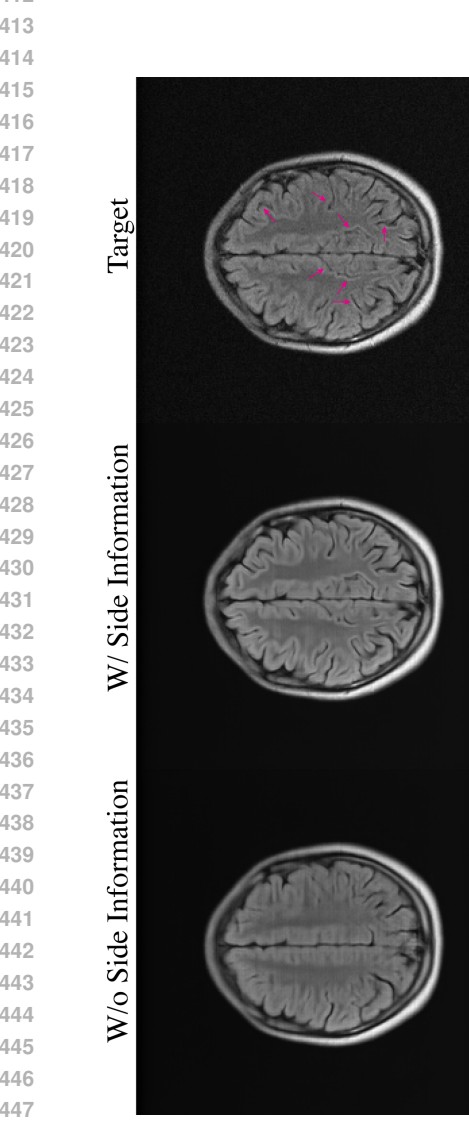

(a) Axial FLAIR image reconstruction without and with side information at 9× acceleration

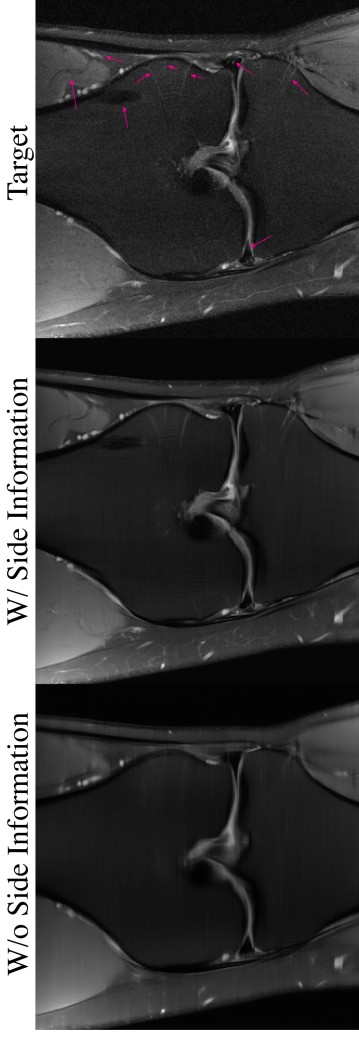

(b) Coronal PDFS image reconstruction without and with side information at 20× acceleration

Figure 13: **Leveraging side information significantly enhances the reconstruction quality. Left:** Reconstructed MR image from highly sparse MR measurements using E2E-VarNet. **Middle:** Reconstructed MR image from the same sparse MR measurements, with additional side information from a different sequence using TGVN (having the same capacity as the E2E-VarNet). **Right:** Ground truth target image, with prominent anatomical features highlighted by purple arrows.

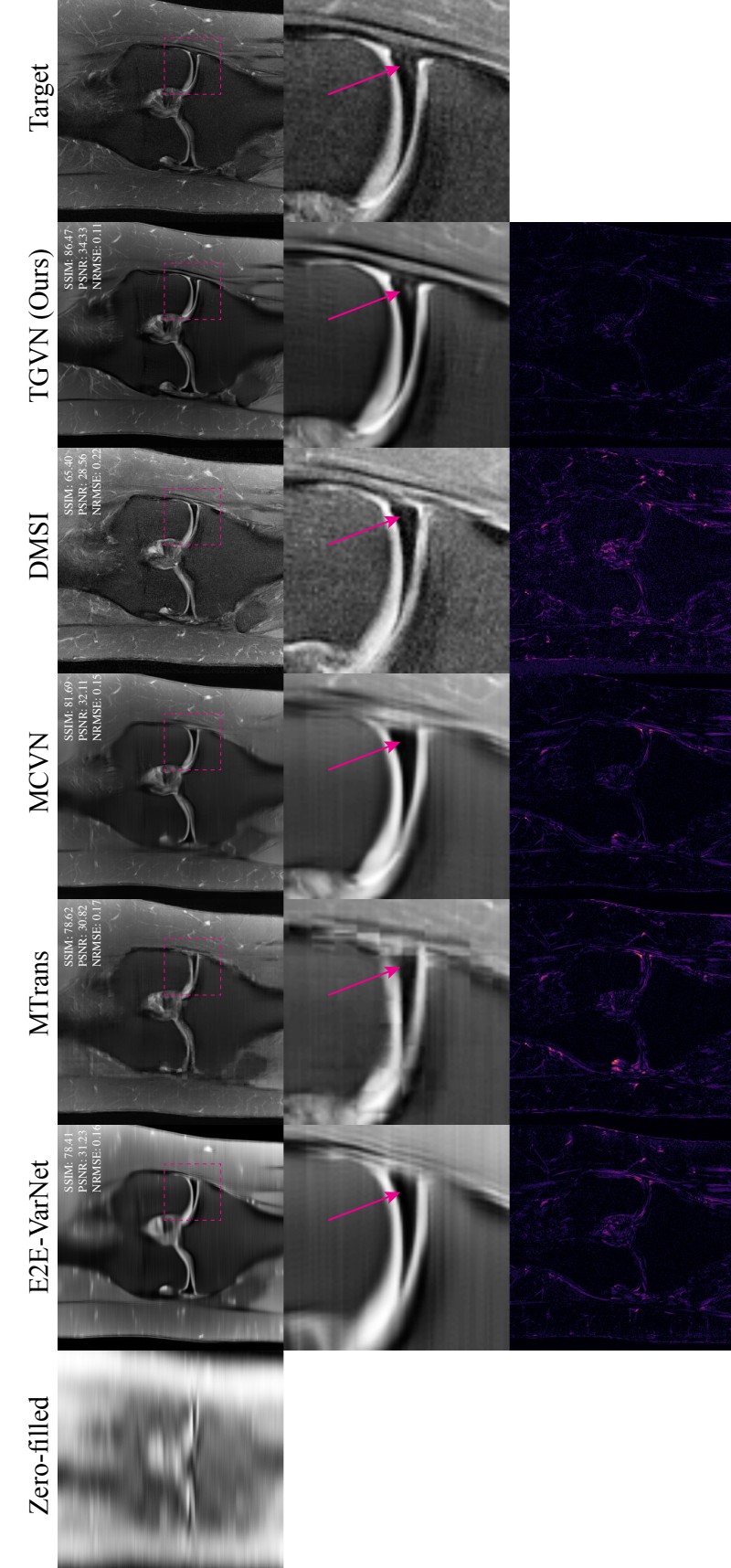

Figure 14: Reconstructions from Set I showing the effectiveness of TGVN in leveraging side information. TGVN is able to reconstruct a high-quality image even at challenging acceleration levels of 20×, in comparison to other baselines. The *meniscus tear*, illustrated in the ground truth image and reconstructions with a purple arrow, is clearly visible in TGVN reconstruction. Top row: Original reconstructions of zero-filled, E2E-VarNet, MTrans, MCVN, DMSI, and TGVN methods, followed by the ground truth image. Middle row: Zoomed-in regions from the upper right corner of the images for better visualization. Bottom row: Absolute differences between each reconstruction and the ground truth, with a consistent color mapping to highlight error magnitudes.

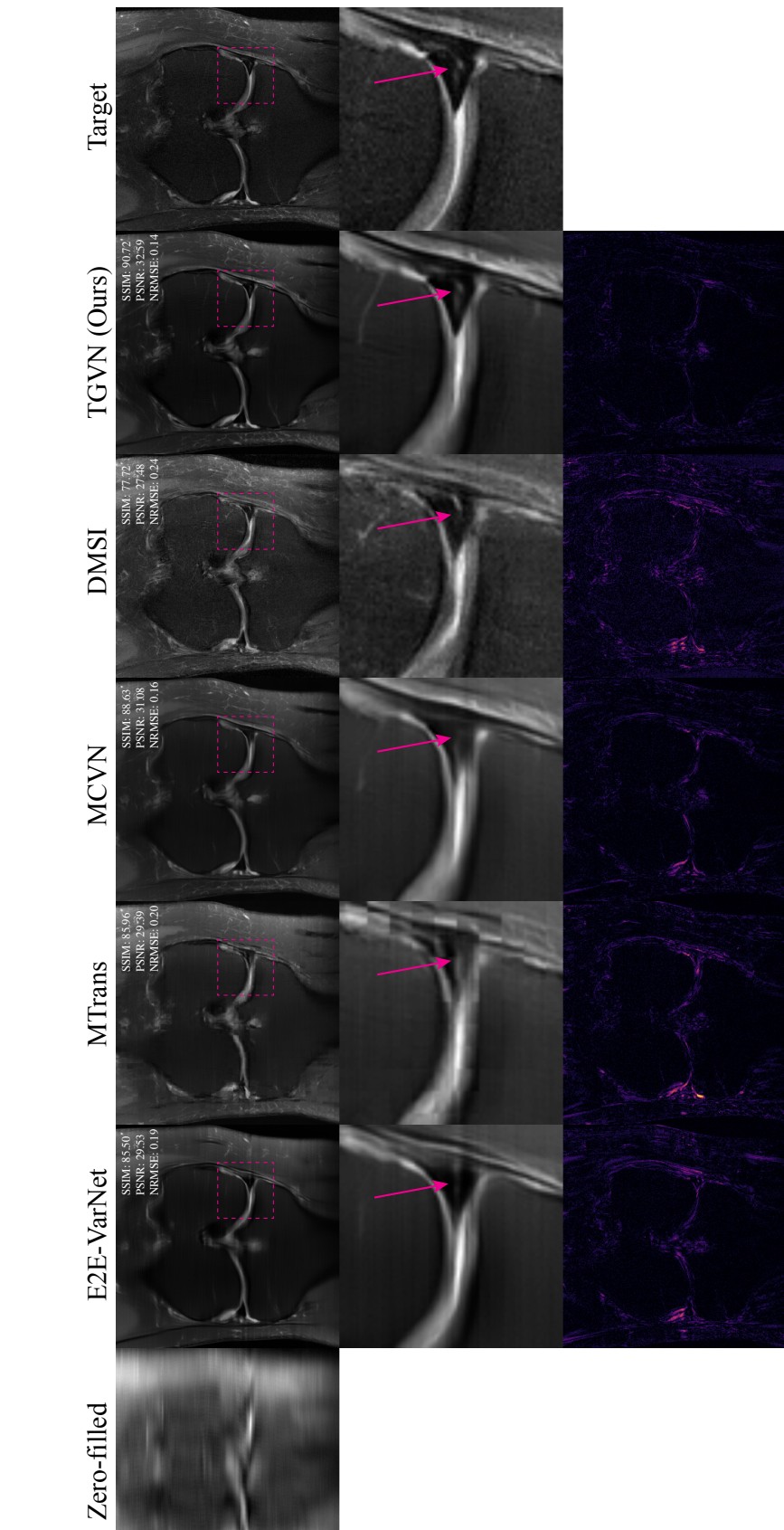

Figure 15: Reconstructions from Set I showing the effectiveness of TGVN in leveraging side information. TGVN is able to reconstruct a high-quality image even at challenging acceleration levels of $20\times$, in comparison to other baselines. The *meniscus tear*, illustrated in the ground truth image and reconstructions with a purple arrow, is clearly visible in TGVN reconstruction. Top row: Original reconstructions of zero-filled, E2E-VarNet, MTrans, MCVN, DMSI, and TGVN methods, followed by the ground truth image. Middle row: Zoomed-in regions from the upper right corner of the images for better visualization. Bottom row: Absolute differences between each reconstruction and the ground truth, with a consistent color mapping to highlight error magnitudes.

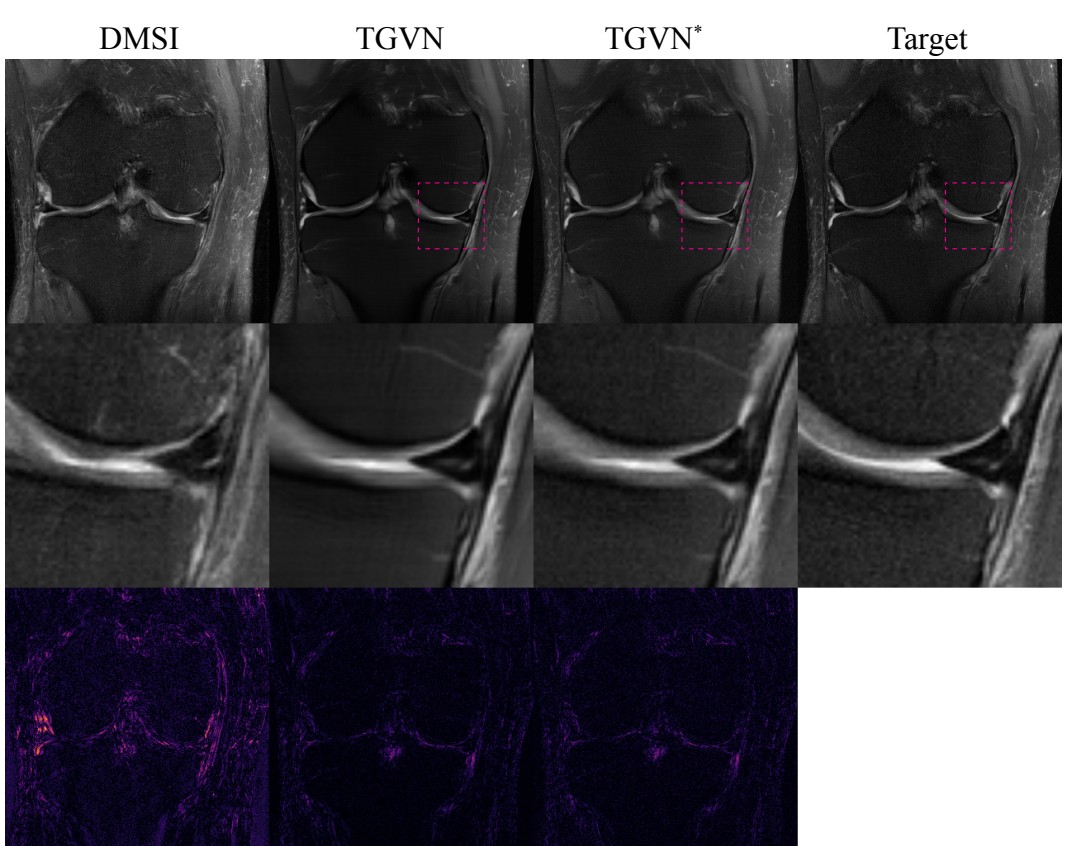

Figure 16: A common challenge in end-to-end reconstructions compared to generative approaches such as DMSI is a residual smoothing of fine image features or background textures. To enhance the subjective perception of sharpness in images, known as acutance in photography, low levels of Gaussian noise were added back to the reconstructed TGVN output, which is represented as TGVN*.

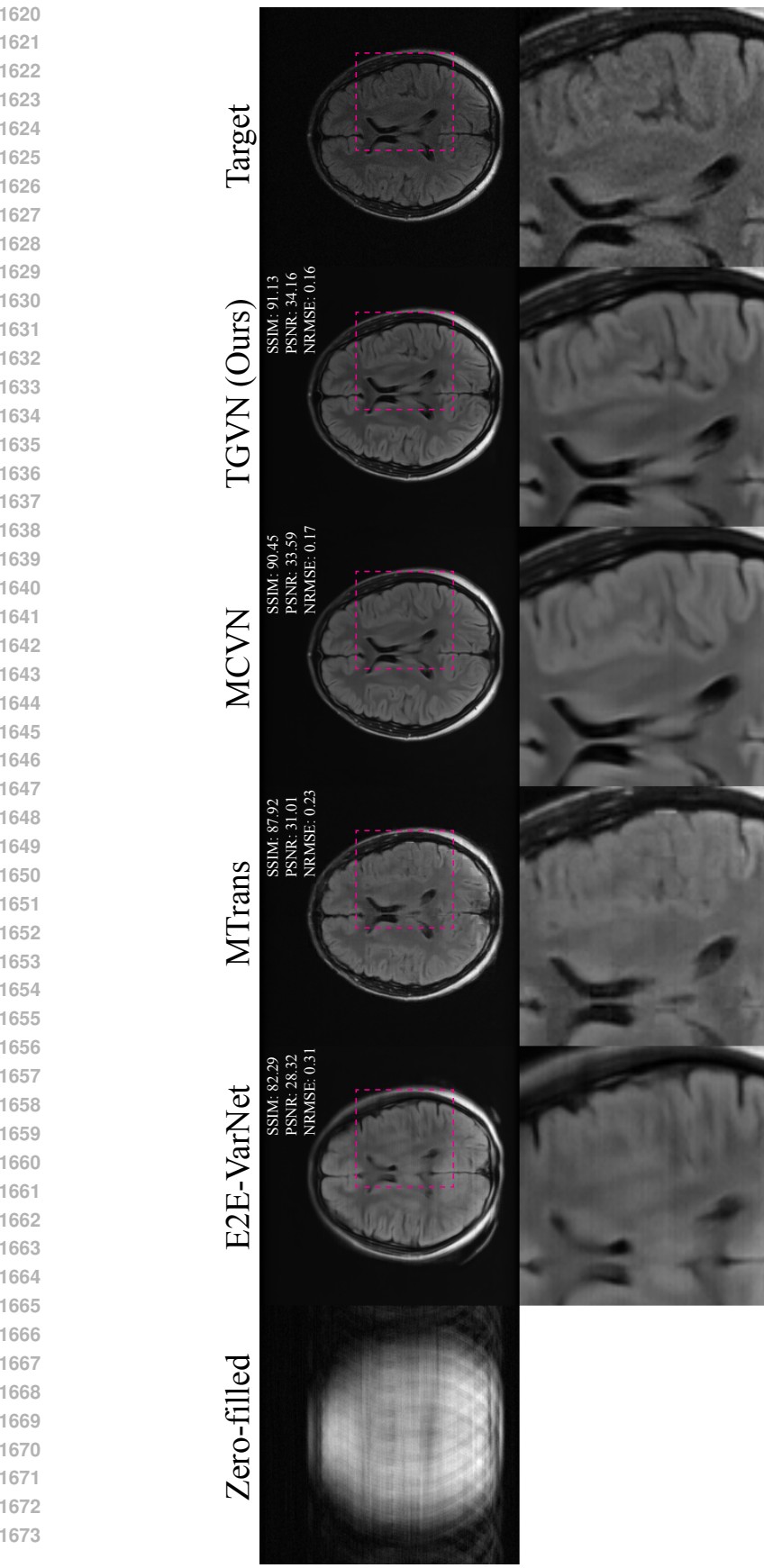

Figure 17: Reconstructions from Set III showing the effectiveness of TGVN at challenging acceleration level of $18\times$ when reconstructing brain images, in comparison to the baselines. Top row: Original reconstructions of zero-filled, E2E-VarNet, MTrans, MCVN, and TGVN methods, followed by the ground truth image. Bottom row: Zoomed-in regions from the upper right corner of each reconstruction and the ground truth, upscaled for better visualization.

