# OpenReview forum: "Leveraging Side Information with Deep Learning for Linear Inverse Problems: Applications to MR Image Reconstruction"
_ICLR.cc/2025/Conference — Submitted to ICLR 2025_

### Official Review · Reviewer_TG2Y · 2024-10-15

**Soundness:** 3
**Presentation:** 2
**Contribution:** 2
**Rating:** 5
**Confidence:** 3

**Summary:**

This paper presents a novel deep learning-based method called Trust-Guided Variational Network. This method addresses the problem of reconstructing high-quality MRI images from undersampled frequency space data. The primary challenge in MRI reconstruction is that undersampling results in ambiguities due to rank-deficient forward operators, leading to multiple non-diagnostic solutions. TGVN integrates relevant side information to constrain the solution space and eliminate undesirable solutions, enhancing image quality.

**Strengths:**

1. This paper incorporates side information into the MRI reconstruction process. It introduces an "ambiguous space consistency constraint" to tackle the ill-posed nature of the problem.
2. The paper is generally clear, with well-organized sections on background, methodology, and empirical validation. The explanations of the core concepts, such as the linear inverse problem and the ambiguous space consistency constraint, are detailed and easy to follow for readers familiar with the domain.

**Weaknesses:**

1. limitation of experiments in terms of the methods that have been compared, please consider comparing some other famous methods such as VN, E2E VarNet.

2. Imposing additional constraints is not a new topic. For example, the paper: Bian, Wanyu, et al. "A learnable variational model for joint multimodal MRI reconstruction and synthesis." is also an optimization-based recon model that utilizes the multicontrast information for enhancing performance. Authors should cite some related paper that leveraging additional contrast.

3. Since it is a model based recon paper, authors need to add optimization based deep learning methods such as ADMM-net, ISTA-net, in the related work. End-to-end and generative methods are not too close to the topic.

4. Authors didn't provide a network structure or whole framework of this work.

**Questions:**

1. From the figure 3 and 12, DMSI provides more details of the tissue and close to the ground truth. The proposed method smooth out some features of the knee joint, fat part.

2. Please indicate the box of the zoom in region in all of the figures.

3. Conduct a doctor study for evaluating the performance of these compared results should strengthen  robustness of the study.

4. In equation 3 and 4, author didn't explain what is \phi.

5. H(s, \gamma) is assumed to be learnable, and s is fixed variable, how does author choose s? Why not updating s iteratively with s^t and x^t alternatively?

---

> ### Author Response · Authors · 2024-11-22
> **Response to Reviewer TG2Y**
>
> We thank the reviewer for their constructive feedback and valuable suggestions, which helped us improve the manuscript.
> Below, we provide our responses to the raised concerns and questions.
>
> **Weaknesses:**
> - limitation of experiments in terms of the methods that have been compared
> > In fact, we compared our method against E2E-VarNet in the submitted version, as shown in Table 1. However, we did not include E2E-VarNet in the figures, as its performance is significantly worse than the methods utilizing side information. Based on your feedback, we have now added E2E-VarNet reconstructions to the figures as well.
>
> - Imposing additional constraints is not a new topic.
> > Thank you for your feedback. Besides this work, we have already referenced more than ten prior studies utilizing side information. However, we missed this particular work and have now mentioned it in our introduction.
>
> - Since it is a model based recon paper, authors need to add optimization based deep learning methods such as ADMM-net, ISTA-net, in the related work
> > In the related work section, we discussed only MR image reconstruction methods that utilize side information. The methods you mentioned are already covered in the introduction, where we provide a broader explanation of MR image reconstruction. We have changed the title of the related work section to *Related Work: Side Information in MR Image Reconstruction* to avoid confusion.
>
> - Authors didn't provide a network structure or whole framework of this work.
> > We have added a figure (Fig. 1 on page 2) illustrating the network structure and how the side information is incorporated into the reconstruction problem using TGVN.
>
> **Questions:**
> 1. From the figure 3 and 12, DMSI provides more details of the tissue and close to the ground truth.
> > Thank you pointing out this issue. A common challenge in end-to-end reconstructions compared to generative approaches such as DMSI is a residual smoothing of fine image features or background textures. To enhance the subjective perception of sharpness in images, known as acutance in photography, low levels of complex-valued Gaussian noise were added back to the reconstructed TGVN output in a new figure in Fig. 16 in Appendix D. We illustrated how this procedure easily changes the acutance. Note that noise levels in DMSI are substantially higher than in ground-truth, which may give the appearance of sharpness but can also obscure key structures, including pathologies.
>
> 2. Please indicate the box of the zoom in region in all of the figures.
> > We have indicated the box for the zoomed-in region as suggested.
>
> 3. Conduct a doctor study for evaluating the performance of these compared results should strengthen robustness of the study.
> > We are certainly interested in validating our results in prospective physician graded studies. However, there is not sufficient time to perform such studies appropriately in this revision.
>
> 4. In equation 3 and 4, author didn't explain what is \phi.
> > On line 178 on page 4 of the original submission, and on line 190 on page 4 of the revised version, $\Phi$ is explained as a *neural network*. If you mean what the exact architecture is, then it is explained in Appendix B, Training and Evaluation Details.
>
> 5. H(s, \gamma) is assumed to be learnable, and s is fixed variable, how does author choose s? Why not updating s iteratively with s^t and x^t alternatively?
> > It is explained, e.g., on line 258 of the original and revised submission,  that $\mathbf{s}$ refers to the side information. The task we are working on involves reconstruction with side information, which refers to utilizing side information to reconstruct the main information. This is why $\mathbf{s}$ is not updated. A related problem is joint multi-contrast reconstruction, where multiple contrasts are reconstructed jointly; however, this is outside the scope of the current work.

---

> ### Author Response · Authors · 2024-12-01
> **Kind Reminder: Reviewer Feedback Deadline**
>
> Dear Reviewer TG2Y,
>
> As the extension period for discussion nears its end, we kindly remind you that we have addressed all your comments and questions in the revised manuscript and provided additional clarifications where needed.
>
> We would greatly appreciate it if you could revisit your evaluations in light of our updates. Please let us know if there is anything further we can clarify before the deadline.
>
> Thank you again for your valuable feedback.
>
> Best regards,
>
> The Authors

---

### Official Review · Reviewer_36te · 2024-11-02

**Soundness:** 3
**Presentation:** 3
**Contribution:** 3
**Rating:** 6
**Confidence:** 4

**Summary:**

This paper proposes a Trust-Guided Variational Network (TGVN), an end-to-end deep learning (DL) method for accelerating MRI reconstruction. Existing physics-based unrolled networks face the problem of ambiguities in the solution space, degrading reconstruction image quality. By introducing the concept of ambiguous space consistency, this work proposes to incorporate side information (i.e., diverse modalities here) into unrolled DL frameworks for further enhancing MRI reconstruction. This approach appears technically and theoretically sound. And extensive experiments confirm the effectiveness of the proposed TGVN model.

**Strengths:**

**Motivation and Novelty**: This work introduces the concept of ambiguous space consistency to address ambiguities in the solution space of existing unrolled methods and proposes using side information to resolve this issue. Although the work largely follows the general framework of unrolled methods, its novelty meets the standards for ICLR conference publication.

**Clarity and organization**: This paper is well-written and easy to follow.

**Experimental evaluation**: This work conducts extensive experiments on two MRI datasets to validate the proposed method, including: 1) comparisons with other unrolled methods without side information and with existing side-information-based methods, and 2) ablation studies on key components of the proposed method. The results confirm the effectiveness of the TGVN model.

**Weaknesses:**

**Clarity**: In this work, ambiguous space consistency is a key concept. Providing some intuitive explanations would improve readability.

**Type of side information**: In this work, different MR image modalities are used as side information. I wonder if the differences between the main information (i.e., measurements) and side information could significantly affect model performance. More specifically, could complementary main and side information (e.g., T1w and T2w) better improve performance? If so, this use of side information appears similar to multi-modality MRI reconstruction. Please clarify the differences between the proposed approach and existing multi-modality MRI reconstruction methods.

**Compared methods**: The proposed method follows a supervised learning paradigm, requiring extensive high-quality MRI images for pre-training. However, state-of-the-art MRI reconstruction methods are not included, such as supervised Transformer-based methods [1][2][3] and unsupervised diffusion methods [4][5].

**Generalization evaluation on different datasets**: On the two datasets, the training, validation, and test are conducted independently. I wonder if the use of side information can enhance the model's generalization to out-of-domain data.

**Under-sampling mask**: The under-sampling masks used in this work are based on Cartesian sampling. Please evaluate the model performance on other types of sampling, such as radial sampling.

**Experimental results**:

1.	I observe that the performance of the E2E VarNet in Exp. 3 is significantly decreased compared to its performance in Exp. 1 and 2. However, other baselines and the proposed method do not show this trend. Please explain this result.

2.	In the ablation studies, only the SSIM metric is reported. Please provide other two metrics (PSNR and NRMSE).

> [1] Guo, Pengfei, et al. "ReconFormer: Accelerated MRI reconstruction using recurrent transformer." IEEE transactions on medical imaging (2023).

> [2] Huang, Jiahao, et al. "Swin transformer for fast MRI." Neurocomputing 493 (2022): 281-304.

> [3] Feng, Chun-Mei, et al. "Task transformer network for joint MRI reconstruction and super-resolution." Medical Image Computing and Computer Assisted Intervention–MICCAI 2021: 24th International Conference, Strasbourg, France, September 27–October 1, 2021, Proceedings, Part VI 24. Springer International Publishing, 2021.

> [4] Chung, Hyungjin, and Jong Chul Ye. "Score-based diffusion models for accelerated MRI." Medical image analysis 80 (2022): 102479.

> [5] Peng C, Guo P, Zhou S K, et al. Towards performant and reliable undersampled MR reconstruction via diffusion model sampling[C]//International Conference on Medical Image Computing and Computer-Assisted Intervention. Cham: Springer Nature Switzerland, 2022: 623-633.

**Questions:**

See the Weaknesses above, please.

---

> ### Author Response · Authors · 2024-11-22
> **Response to Reviewer 36te**
>
> We thank the reviewer for their constructive feedback and valuable suggestions, which helped us improve the manuscript.
> Below, we provide our responses to the raised concerns and questions.
>
> **Weaknesses:**
> - Clarity
> > Based on your feedback, we have expanded Sec. 4 explaining our novel work. We have also added a figure on page 2 (Fig. 1) to explain our architecture along with the general framework.
>
> - Type of side information
> > In the original manuscript, as well as the revised version, we have included a brief discussion of complementary information at the beginning of Sec. 4. As long as the side information $\mathbf{s}$ is conditionally dependent on the image being sought ($\mathbf{x}$) given the under-sampled measurements $\mathbf{k}$, we can exploit this side information in resolving ambiguities. In the literature, we have observed that multi-contrast MRI is sometimes referred to as multi-modal MRI. However, functional MRI, diffusion MRI, and MR spectroscopy are occasionally also referred to as multi-modal MRI. Our work is generic enough to incorporate either, but the datasets we use contain only multi-contrast data. To avoid confusion, we have opted to use the term multi-contrast instead of multi-modal. Additionally, joint multi-contrast MRI reconstruction is sometimes referred to as multi-modal MRI reconstruction. While our work can be adapted to perform joint reconstruction, it is not the focus of this manuscript.
>
> - Compared methods
> > Based on your feedback, we have discussed these single-contrast methods in the introduction section. However, since the main information is accelerated at levels up to $20\times$, the use of side information is crucial in our experiments. All of the suggested methods, on the other hand, are designed for single-contrast reconstruction and perform similarly to E2E-VarNet, which does not exploit side information. For clarification, we want to note that the baseline MTrans is transformer-based, and the baseline DMSI is diffusion-based, both utilizing side information.
>
> - Generalization evaluation on different datasets
> > This is an interesting point that we have not explored yet. Considering the time required for each experiment, (up to two weeks for each knee experiment), we have not been able to test this approach in time of the current submission. It will guide our future research direction.
>
> - Under-sampling mask
> > As Cartesian under-sampling is by far the most frequently used in practice, we have focused on the Cartesian case. We fully expect our method to work with other sampling trajectories; however, nontrivial modifications would be required, which would take longer than the time available for this revision.
>
> - Experimental results 1
> > Thank you for this insightful question. Exp. 3 was performed using the M4Raw dataset, which was acquired at $0.3$T with $4$ coils and has a small matrix size and low SNR. In contrast, Exp. 1 and 2 were performed using the fastMRI dataset, acquired at $1.5$T and $3$T with $15$ coils and a much larger matrix size and higher SNR. Considering these factors, achieving high acceleration without side information is significantly more challenging in Exp. 3, compared to the first two experiments. Furthermore, E2E-VarNet is the only method that does not use the side information, which demonstrates how important side information becomes in challenging cases.
>
> - Experimental results 2
> > We have added PSNR and NRMSE scatter plots in the ablation studies. Furthermore, we have significantly expanded our ablation studies and Appendices.

---

> > ### Comment · Reviewer_36te · 2024-11-22
> >
> > Thank you for your thoughtful responses, which address many of my concerns. However, regarding the evaluation of model generalization on unseen datasets, you mentioned, "up to two weeks for each knee experiment." I would like to clarify whether I understood correctly: why does testing the pre-trained TGVN model on an unseen dataset require such a long time? Here I just wonder if using side information could enhance the model generalization to out-of-domain data, rather than requiring the model to be trained from scratch.

---

> ### Author Response · Authors · 2024-11-24
> **Generalization to out-of-domain data**
>
> Dear reviewer,
>
> Previously, we interpreted your point differently. For E2E-VarNet and TGVN of the same capacity, which were trained to reconstruct 20x under-sampled main information (knee PDFS), we performed inference on the M4Raw brain FLAIR test split. The statistics are provided in the table below:
>
> | Metric | E2E-VarNet        | TGVN           |
> |--------|---------------|----------------|
> | SSIM   | 42.08±0.17    | 55.66±0.24     |
> | PSNR   | 10.68±0.05    | 19.17±0.06     |
> | NRMSE  | 1.56±0.004    | 0.59±0.004     |
>
> While side information appears to enhance generalization by significantly improving the metrics, we would like to emphasize that the image quality degrades substantially (cf. Table 1 in the manuscript) and the reconstructions remain far from being diagnostic and are not acceptable in this generalization experiment, even with side information.

---

> > ### Comment · Reviewer_36te · 2024-11-25
> >
> > Thank you for your additional results. The evaluation of OOD data shows that side information can effectively improve model generalization (about 9 dB in PSNR). Although the performance decreased significantly, it may offer a promising solution for improved generalization. Again, thanks for your prompt reply. I have also reviewed the comments from other reviewers. I decide to keep my original rating of 6.

---

> > > ### Author Response · Authors · 2024-11-25
> > >
> > > Thank you for your feedback and for taking the time to review our additional results. We appreciate your thoughtful consideration of our work.

---

> > > ### Author Response · Authors · 2024-11-29
> > >
> > > Dear Reviewer,
> > >
> > > Thank you for engaging with our rebuttal. At the time of your response, no other reviewers had provided feedback, and you mentioned having reviewed the comments from others. While two reviewers have not yet responded, we wanted to share that **Reviewer GU7v** has since acknowledged the improvements and updated their rating from **3 to 6**.
> > >
> > > We kindly ask if you might consider re-evaluating your own rating in light of this. We greatly value your time and thoughtful feedback.
> > >
> > > Best regards,
> > >
> > > The Authors

---

### Official Review · Reviewer_hAQd · 2024-11-02

**Soundness:** 2
**Presentation:** 1
**Contribution:** 2
**Rating:** 3
**Confidence:** 3

**Summary:**

This paper deals with solving the inverse problem in magnetic resonance imaging by leveraging domain knowledge. Therefore, the authors introduce the so-called Trust-Guided Variational Network (TGVN).  Despite the data consistency where the reconstructed images should align with the measurements the ambiguous space consistency is included which ensures that the projected data aligns with the side information.

**Strengths:**

* The idea to incorporate side information is good. Reducing acquisition time and therefore gathering undersampled data which leads to artifacts in the reconstruction process will require sophisticated but trustable reconstruction algorithms as MRI is a versatile application.
* “Easy” integration of the ambiguous space consistency constraint.
* The approach is end-to-end trainable.

**Weaknesses:**

* There exist already many attempts to incorporate side information.
* The paper includes a lot of general information in the paper but a quite short explanation of the novel work.
* In this paper, four research questions are mentioned but only two are discussed whereby also the other ones, especially Q3, would be also considered as important as it is part of the main contribution of this paper. The general stuff should be shortened and a focus of the new method and the results should be
* The authors advertise that different types of side information can be included but not exactly how. Furthermore, it is mentioned that this is part of future work
* The generalizability is not shown across different anatomies, as stated. The brain and knee dataset are handled separately.
* A lot of stuff is described generally but the key points are missing.

**Questions:**

* Why should we use this approach and not another one to incorporate side information? What is the benefit of this method (beyond more feasible solutions which is the goal when incorporating side information)?
* How does the threshold effect the results?
* Which projection was chosen?
* How is the module learned?

---

> ### Author Response · Authors · 2024-11-22
> **Response to the Reviewer hAQd**
>
> We thank the reviewer for their constructive feedback and valuable suggestions, which helped us improve the manuscript.
> Below, we provide our responses to the raised concerns and questions.
>
> **Weaknesses:**
> - There exist already many attempts to incorporate side information.
> > While there have been some attempts to incorporate side information, to the best of our knowledge, no previous approach has investigated an ambiguous space framework with end-to-end trainable variational networks. Furthermore, we have achieved statistically significant improvements over a variety of baselines utilizing side information. In fact, we have not seen acceleration levels as high as those demonstrated in this work using other side information approaches.
>
> - The paper includes a lot of general information in the paper but a quite short explanation of the novel work.
> > Based on your suggestion, we shortened the literature review and extended the explanation of the novel work in Sec. 4. Furthermore, we have illustrated the general framework, a TGVN cascade, and a full TGVN on page 2 (Fig. 1).
>
> - In this paper, four research questions are mentioned but only two are discussed whereby also the other ones, especially Q3, would be also considered as important as it is part of the main contribution of this paper.
> > Due to page limitations, two research questions were answered in the main text while the other two were discussed in the Appendix. Essentially, our experiments demonstrated that **(I)** The use of side information significantly improves the reconstruction quality. **(II)** TGVN is more effective in leveraging side information than the relevant DL baselines of the same capacity which also utilize side information. **(III)** Projection onto the ambiguous space improves the reconstruction quality significantly. **(IV)** TGVN is robust to moderate under-sampling or small misregistration of the side information.
>
> - The authors advertise that different types of side information can be included but not exactly how. Furthermore, it is mentioned that this is part of future work
> > In our proposed architecture, the side information is utilized in the $H$ block. Hence, different types of side information can be incorporated by modifying the $H$ block. For instance, if the side information comes from other imaging modalities such as CT, ultrasound, or PET, the same architecture can be used. For non-image data, such as textual or tabular information, embeddings obtained from a pre-trained, frozen encoder can be used to train a decoder end-to-end, along with all the other parameters of TGVN. In this case, the $H$ block will function as a decoder that maps features to image space.
>
> - The generalizability is not shown across different anatomies, as stated. The brain and knee dataset are handled separately.
> > Our intention was not to claim that one network can be trained and used for all anatomies. Rather, we intended to convey that the approach can work across various anatomies. Based on your feedback, we have changed the wording from generalizability to robustness.
>
> - A lot of stuff is described generally but the key points are missing.
> > We have addressed this general concern to the best of our ability, by shortening the introductory text and expanding the descriptions of the methods, results, and Appendices.
>
> **Questions:**
> - Why should we use this approach and not another one to incorporate side information?
> > With our method, we have been able to achieve acceleration levels substantially higher than what has been feasible using other approaches. Furthermore, our approach is designed to minimize the risk of hallucinations while effectively leveraging side information to maximize the desired similarity metric between the reconstructions and reference images. The main experiments we presented demonstrate the effectiveness of TGVN compared to relevant deep learning baselines of the same capacity (which also utilize side information).
>
> - How does the threshold effect the results?
> > As the threshold gets smaller, the ambiguous space converges to the null space. Since the null space is trivial for acceleration rates lower than the number of coils in parallel imaging, this means the optimal threshold will likely be large. In our experiments, we in fact learned the threshold as we have used the approximate projection explained in Sec. 4.2 in the original as well as in the revised submission.
>
> - Which projection was chosen?
> > The approximate projection was chosen due to computational complexity of using the exact projection. This was noted in Appendix B of the original submission as well as of the revised version.
>
> - How is the module learned?
> > The module is learned end-to-end with all the other parameters of TGVN, as we explained in Sec. 4.2. Specifically, $\Omega$ is selected to maximize the multi-scale structural similarity (MS-SSIM) between the reconstructions and the reference images.

---

> ### Author Response · Authors · 2024-12-01
> **Kind Reminder: Reviewer Feedback Deadline**
>
> Dear Reviewer hAQd,
>
> As the extension period for discussion nears its end, we kindly remind you that we have addressed all your comments and questions in the revised manuscript and provided additional clarifications where needed.
>
> We would greatly appreciate it if you could revisit your evaluations in light of our updates. Please let us know if there is anything further we can clarify before the deadline.
>
> Thank you again for your valuable feedback.
>
> Best regards,
>
> The Authors

---

### Official Review · Reviewer_GU7v · 2024-11-04

**Soundness:** 3
**Presentation:** 3
**Contribution:** 2
**Rating:** 6
**Confidence:** 4

**Summary:**

The paper with title: Leveraging Side Information with Deep Learning for Linear Inverse Problems: Applications to MR Image Reconstruction presents a Deep learning-based framework that can leverage the information from other contrast to help the reconstruction, the key framework uses a Null space projection method to disentangle the side information. Results on fastMRI dataset show improved image quality compared to other SOTA methods.

**Strengths:**

1. Using prior knowledge from other contrast is a very interesting topic in terms of MRI reconstruction since for clinical routine, usually multiple contrasts are acquired.

2. Using Null-space to disentangle the contrast-information is interesting.

**Weaknesses:**

1. The presentation and story telling is not clear and there isn't any figure illustration of the problem/method/novelty, which makes the paper very difficult to follow, especially the null-space and theory part.

2. Using multi-contrast information for joint reconstruction has been a popular topic, but the authors didn't compare with any-of-them.

3. Following the previous weaknes, instead of using Null-space, one way to extract information from side contrast is by learning an representation, you should incorporate this experiments - like for example in E2E experiment, try inputing two channels.

4. Instead of using aliased side information (3x undersampled image from side contrast), it makes sense to use a reconstructed version of other contrast, like using E2E, have you done this experiments?

5. ~20x undersmapling doesn't make sense at all, in practice, or in fastMRI dataset, only a few lines are sampled beside the ARC region, how do you expect recovering any details? especially for pathologies.

**Questions:**

1. Do you use paired data for training - like the main and side contrasts from the same object/ same anatomy? how do you do the alignment?

2. I need more ablation study to understand how the side information contributes.

---

> ### Author Response · Authors · 2024-11-22
> **Response to the Reviewer GU7v**
>
> We thank the reviewer for their constructive feedback and valuable suggestions, which helped us improve the manuscript.
> Below, we provide our responses to the raised concerns and questions.
>
> **Weaknesses:**
> 1. The presentation and story telling is not clear and there isn't any figure illustration of the problem/method/novelty, which makes the paper very difficult to follow, especially the null-space and theory part.
> > We have added a figure illustrating the general framework, a TGVN cascade, and a full TGVN on page 2 (Fig. 1).
>
> 2. Using multi-contrast information for joint reconstruction has been a popular topic, but the authors didn't compare with any-of-them.
> > We acknowledge that using multi-contrast information for joint reconstruction has been a popular topic. In the original submission, we compared our method against three baselines that also utilize side information. These baselines can also be applied in a joint reconstruction setting; however, their performance deteriorates in this context. Specifically, the end-to-end baselines process both contrasts and output two reconstructions. In joint reconstruction, the network parameters are optimized to minimize the sum of losses from both contrast reconstructions and their respective references. In contrast, in reconstruction with side information, the parameters are optimized to minimize the loss between only the target contrast and the reference image. Consequently, the baselines we provided are more likely to be stronger than the same baselines trained in a joint reconstruction setting in terms of target contrast reconstruction performance. Besides all this, the diffusion baseline (DMSI) is applied in the joint reconstruction setting, as the side information in the knee experiments is $2\times$ or $3\times$ under-sampled while the score models are trained with fully sampled images.
>
> 3. Following the previous weaknes, instead of using Null-space, one way to extract information from side contrast is by learning an representation
> > In the original submission, as well as in the revised version, ablation study A.1 in Appendix investigates the effect of projection. In this experiment, we utilized the side information without applying the ambiguous space projection by learning representations and incorporating them into the E2E-VarNet architecture. Hence, we have presented this result in the original manuscript and significantly expanded them in the revised version to make this clearer.
>
> 4. Instead of using aliased side information ($3\times$ undersampled image from side contrast), it makes sense to use a reconstructed version of other contrast, like using E2E, have you done this experiments?
> > We have done this experiment but did not present it in the original submission. Following your suggestions, we have appended it to Appendix A.2.2. While reconstructed side information might seem intuitive, we observed no statistically significant improvements in reconstruction metrics compared to a single-stage approach that directly utilizes the under-sampled side information.
>
> 5. ~20x undersmapling doesn't make sense at all, in practice, or in fastMRI dataset, only a few lines are sampled beside the ARC region, how do you expect recovering any details? especially for pathologies.
> > Without any side information, there is no hope of recovering details at these acceleration levels. However, the side information in our experiments is only moderately under-sampled (at most $3\times$), providing significant anatomical and potentially pathological detail, as demonstrated by the clear depictions of meniscal tears in our results. Additionally, in our experiments, we used smaller ARC regions than usual, as explained in Sec. 5 in our manuscript. We are ready to make the code publicly available. Since the datasets are already accessible, our results can be reproduced easily.
>
> **Questions:**
> 1. Do you use paired data for training - like the main and side contrasts from the same object/ same anatomy? how do you do the alignment?
> > We used the fastMRI and M4Raw datasets, which are paired but occasionally contain small misregistrations. Additionally, we performed an ablation study (Appendix A.2.1) to investigate the effect of deliberately introducing misregistrations. The results indicated that small misregistrations can be tolerated if training also introduces such random misregistrations, and reconstruction with misregistered side information still performs significantly better than reconstruction without any side information.
>
> 2. I need more ablation study to understand how the side information contributes.
> > Based on your suggestion, we have added a comparison between two-stage and single-stage (proposed) reconstruction, presented in Appendix A.2.2. Furthermore, we significantly expanded the other ablation studies to provide a more comprehensive understanding of the impact of side information on reconstruction performance.

---

> > ### Comment · Reviewer_GU7v · 2024-11-25
> > **Great response!**
> >
> > I sincerely appreciate the authors feedbacks which address quite portion of my concerns. the paper looks indeed much more polished.
> > I updated my score.
> > One question, you mentioned that using recontructed side information as input barely improve the results - no significant differences. whats the intuiation behind it, if the input is aliased, how do the networks learn the side information, this might relate to the network design (like whether using transformer block or so).

---

> > > ### Author Response · Authors · 2024-11-25
> > >
> > > **Dear Reviewer,**
> > >
> > > Thank you for carefully reviewing our revised manuscript. We greatly appreciate your feedback and the update.
> > >
> > > The question of why reconstructed side information does not necessarily improve results in a statistically significant way is an **excellent one**. We are currently using U-nets in the $\mathcal{H}$ blocks to process the side information, but a transformer architecture could work just as well. Below, we present our intuition by comparing mutual information upper bounds for each approach. We would like to note that this is not a rigorous proof but rather a heuristic explanation that we believe is insightful.  As described in the paper, let $\mathbf{s}$ denote the moderately under-sampled side information, with the corresponding fully sampled side information denoted by $\widetilde{\mathbf{s}}$. Let $\mathbf{k}$ denote the heavily under-sampled main information, and let $\mathbf{x}$ denote the image being sought, i.e., the ground-truth image corresponding to the fully-sampled main information.
> > >
> > > The notion of **conditional mutual information** can be used to quantify the potential of utilizing side information: $I(\mathbf{x};\mathbf{s}|\mathbf{k})$ represents the reduction in the uncertainty of $\mathbf{x}$ due to the knowledge of $\mathbf{s}$ when $\mathbf{k}$ is given. This quantity serves as an upper bound: a well-designed method operating near this limit will perform well, while a poor design will yield almost no reduction in uncertainty. Clearly, fully sampled side information provides greater upper bound, which can be expressed as $I(\mathbf{x};\widetilde{\mathbf{s}}|\mathbf{k}) \geq I(\mathbf{x};\mathbf{s}|\mathbf{k})$. If a method is operating close to the mutual information limit, then one would expect better reconstructions when using the fully sampled side information $\widetilde{\mathbf{s}}$.
> > >
> > > In a two-stage approach, a network $g$ is first learned, parametrized by $\tau$, such that $\mathcal{L}(g(\mathbf{s};\tau), \widetilde{\mathbf{s}})$ is minimized. Once the optimal parameters $\widehat{\tau}$ are learned, the second stage involves reconstructing the main information. The potential of utilizing side information in this two-stage approach can be expressed as $I(\mathbf{x};g(\mathbf{s};\widehat{\tau})|\mathbf{k})$.
> > >
> > > The critical insight here lies in the use of the **data processing inequality (DPI)**, which states that $I(\mathbf{x};g(\mathbf{s};\widehat{\tau})|\mathbf{k}) \leq I(\mathbf{x};\mathbf{s}|\mathbf{k})$, since $g$ is a deterministic function of $\mathbf{s}$. Therefore, if the single-stage method is designed to be near-optimal, such that it maximally exploits the potential $I(\mathbf{x};\mathbf{s}|\mathbf{k})$, a two-stage approach **might not offer additional benefits**. This follows directly from the inequality $I(\mathbf{x};\mathbf{s}|\mathbf{k}) \geq I(\mathbf{x};g(\mathbf{s};\widehat{\tau})|\mathbf{k})$ imposed by the DPI.
> > >
> > > We hope you find this intuition helpful.
> > >
> > > **Best regards,**
> > > The authors

---

> > > > ### Comment · Reviewer_GU7v · 2024-11-27
> > > >
> > > > this makes sense,
> > > > whats the perceptive field of your U-Net, if the perceptive field is very small, then we don't expect the U-Net to learn long-range corresponse that happened in aliased undersampled (zero-filled recon), thats why I brought up using transformer-based network.

---

> > > > > ### Author Response · Authors · 2024-11-27
> > > > >
> > > > > The receptive field of each U-net is $200\times 200$. Considering that the matrix sizes for fastMRI and M4Raw are $320\times 320$ and $256\times 256$, respectively, we believe that increasing the receptive field further might slightly enhance the reconstruction quality. However, we do not anticipate significant improvements from such an increase. We would appreciate your thoughts on this hypothesis.

---

### Author Response · Authors · 2024-11-24

Dear Reviewers,

We hope this message finds you well.

We wanted to kindly remind you that we have carefully addressed all your comments and questions in the revised manuscript and provided additional clarifications where needed. As the discussion period is nearing its end, we would greatly appreciate it if you could revisit your evaluations in light of the responses and updated information we have shared.We are sincerely grateful for your time and thoughtful feedback throughout this process, and we hope that our efforts to address your concerns are reflected in your final review. Please don’t hesitate to let us know if there is anything further we can clarify or improve.

Thank you again for your invaluable insights and contributions.

Best regards,
The authors

---

### Meta-Review · Area_Chair_HAsL · 2024-12-17

**Metareview:**

The paper presents an end-to-end deep learning-based method, called that reliably incorporates side information into LIPs to eliminate undesirable solutions from the ambiguous space of the forward operator. However, the reviewers believe that imitation of experiments in terms of the methods that have been compared. The description of the relevant work is not very appropriate, while optimization based deep learning methods should be added. More intuitive explanations about the key concept should be provided.

Based on the average rating of the reviewers, I believe the paper is not yet ready for publication at ICLR.

**Additional Comments On Reviewer Discussion:**

Reviewer GU7v is satisfied with the authors’ response and raise the score, but the overall contribution does not seem substantial for publication at ICLR.

---

### Decision · Program_Chairs · 2025-01-22

Reject